# Investigating Redundancy in Multimodal Large Language Models with Multiple Vision Encoders

**Yizhou Wang**[1,2,*,†], **Song Mao**[1,*], **Yang Chen**[1,3,*,†], **Yufan Shen**[1], **Yinqiao Yan**[4],
**Pinlong Cai**[1], **Ding Wang**[1], **Guohang Yan**[1], **Zhi Yu**[3], **Xuming Hu**[2], **Botian Shi**[1]
[1]Shanghai Artificial Intelligence Laboratory
[2]The Hong Kong University of Science and Technology (Guangzhou)
[3]Zhejiang University, [4]Beijing University of Technology
{wangyizhou1, maosong, chenyang3}@pjlab.org.cn

## Abstract

Recent multimodal large language models (MLLMs) increasingly integrate multiple vision encoders to improve performance on various benchmarks, assuming that diverse pretraining objectives yield complementary visual signals. However, we show this assumption often fails in practice. Through systematic encoder masking across representative multi-encoder MLLMs, we find that performance typically degrades gracefully—and sometimes even improves—when selected encoders are masked, revealing pervasive encoder redundancy. To quantify this effect, we introduce two principled metrics: the **Conditional Utilization Rate (CUR)**, which measures an encoder's marginal contribution in the presence of others, and the **Information Gap (IG)**, which captures heterogeneity in encoder utility within a model. Using these tools, we observe: (i) strong specialization on tasks like OCR & Chart, where a single encoder can dominate with a CUR $> 90\%$, (ii) high redundancy on general VQA and knowledge-based tasks, where encoders are largely interchangeable, (iii) instances of detrimental encoders with negative CUR. Notably, masking specific encoders can yield up to $16\%$ higher accuracy on a specific task category and $3.6\%$ overall performance boost compared to the full model. Furthermore, single- and dual- encoder variants recover over $90\%$ of baseline on most non-OCR tasks with substantially lower training resources and inference latency. Our analysis challenges the "more encoders are better" heuristic in MLLMs and provides actionable diagnostics for developing more efficient and effective multimodal architectures. The project website is available at https://github.com/MaoSong2022/Encoder-Redundancy.

## 1 Introduction

Multimodal large language models (MLLMs) have marked a major leap in artificial intelligence (AI), exhibiting remarkable prowess in integrating visual and textual information for complex generation and reasoning tasks (OpenAI, 2025a; DeepMind, 2025; Anthropic, 2024; Bai et al., 2025; Zhu et al., 2025). Their ability to interpret images (Luo et al., 2024), answer visual questions (Zhu et al., 2025; Li et al., 2025a), and perform visual reasoning (OpenAI, 2025b; Peng et al., 2025) has positioned them at the forefront of AI research.

A prominent architectural trend for enhancing visual capabilities of MLLMs is the incorporation of multiple, distinct vision encoders. The rationale is intuitive: different encoders, pre-trained with varied objectives or architectures, could capture complementary aspects of vision—spanning global semantics (Radford et al., 2021; Zhai et al., 2023) to fine-grained pixel-level details (Oquab et al., 2023; Kirillov et al., 2023), thereby providing a richer representation to the language model (Tong et al., 2024b; Lu et al., 2024a; Jiang et al., 2024; Shi et al., 2024; Tong et al., 2024a; Li et al., 2024).

---

* Equal contribution, † This work was done during an internship at Shanghai AI Laboratory.

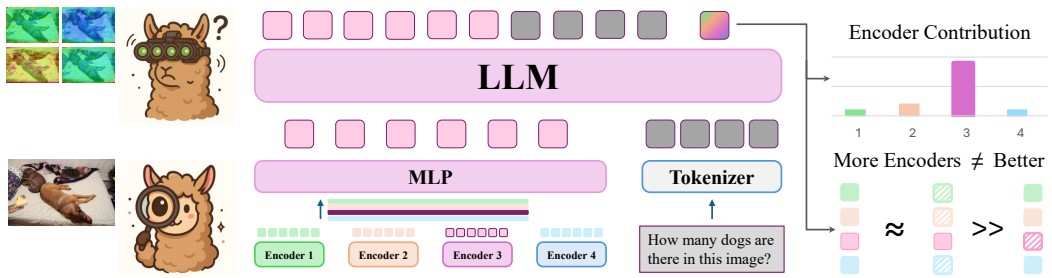

Figure 1: **An illustration of encoder redundancy**. Different vision encoders provide similar or conflict visual cues, by ablating one or several of them the performance maintain or even improve.

However, the assumption that *more encoders are always better* is increasingly being challenged. Emerging evidence suggests that performance gains from additional encoders are often marginal, and in some cases, multi-encoder models even underperform their counterparts with fewer (Shi et al., 2024; Fan et al., 2024). This counterintuitive outcome reveals a critical, yet underexplored issue: *encoder redundancy*. Such redundancy occurs when encoders provide overlapping or conflicting cues, leading to fusion difficulties, distraction from irrelevant signals, and inefficient use of computational resources. Figure 1 illustrates the encoder redundancy phenomenon, where a multi-encoder MLLM strongly depends on a specific encoder to accomplish a domain-specific task.

While significant research has focused on designing sophisticated fusion mechanisms for multi-encoder MLLMs (Kar et al., 2024; Wang et al., 2025; Shen et al., 2024), the fundamental question of whether and to what extent each encoder provides unique, non-redundant information remains largely unexplored. To address this gap, we first empirically demonstrate the presence of redundancy by systematically masking individual vision encoders in representative multi-encoder MLLMs (e.g., Eagle (Shi et al., 2024)) and measuring the impact on performance across a wide range of benchmarks (Tong et al., 2024a). Second, we introduce the **Conditional Utilization Rate (CUR)**, which quantifies the marginal contribution of each encoder given the presence of others. A low or negative CUR indicates that an encoder is redundant or even detrimental. Building on this, we define the **Information Gap (IG)** as the difference between the maximum and minimum CUR values, capturing the disparity in encoder contributions. A large IG signifies a poorly balanced encoder set, with some encoders dominating and others underutilized. Finally, we analyze factors such as LLM size and encoder type that influence redundancy.

Our experiments confirm that significant redundancy is prevalent in modern multi-encoder MLLMs. For instance, a two-encoder subset of Cambrian-1 8B surpasses the full model by $1.7\%$, while masking two encoders of Eagle-X5 7B retains $96\%$ of its five-encoder baseline performance. Beyond accuracy, redundancy reduction also improves efficiency: in our setup, training a dual-encoder variant is $1.53\times$ faster than its five-encoder counterpart while retaining $94\%$ performance. Together, these results show that multi-encoder MLLMs often carry substantial redundancy: adding more vision encoders to MLLMs often yields severely diminishing returns, resulting in performance gains that are negligible compared to the substantial increase in required training resources and inference latency. CUR and IG provide actionable diagnostics for designing more efficient and effective MLLMs by balancing training cost and performance.

In summary, this paper makes the following contributions:

1. We provide the first systematic, quantitative study confirming encoder redundancy and practical inefficiency in multi-encoder MLLMs.

2. We introduce CUR (Conditional Utilization Rate) and IG (Information Gap) as principled, model-agnostic metrics to quantify encoder utility and imbalance.

3. We demonstrate that dual-encoder variants achieve $> 90\%$ of full performance at significantly reduced computational cost, offering an actionable framework for efficient MLLM design.

## 2 RELATED WORK

### 2.1 MULTIMODAL LARGE LANGUAGE MODELS

The landscape of MLLMs has evolved rapidly, with models demonstrating increasingly sophisticated capabilities for understanding and generating multimodal content. Early influential work like Flamingo pioneered the integration of pre-trained vision encoders and Large Language Models (LLMs) by introducing mechanisms like resamplers for token reduction and cross-attention layers for feature fusion (Alayrac et al., 2022). Subsequently, LLaVA presented a simpler yet effective architecture consisting of a vision encoder, an LLM, and a projection layer, establishing a modular paradigm that facilitated scalability and adaptation (Liu et al., 2024). Efforts to enhance visual processing have included mPLUG's visual abstractor for handling high-resolution inputs (Li et al., 2022) and InternVL's dynamic aspect ratio matching (Chen et al., 2024). Similarly, Qwen-VL series introduced techniques like 2D-RoPE and M-RoPE to better model inter-modal relationships (Wang et al., 2024a; Bai et al., 2025). These advancements underscore a continuous drive towards richer visual understanding and more effective vision-language alignment in MLLMs.

### 2.2 EMPLOYING MULTIPLE VISION ENCODERS IN MLLMS

Our research directly engages with the growing body of work on multi-encoder MLLMs. The primary motivation behind multi-encoder MLLMs architectures is to harness diverse visual features by combining encoders pre-trained with different objectives or on varied data. For instance, DeepSeek-VL (Lu et al., 2024a) integrates SigLIP (Zhai et al., 2023) for semantic understanding and SAM-B (Kirillov et al., 2023) for visual grounding. HiLight (Wang et al., 2024b), Mini-Gemini (Li et al., 2024), and CogAgent (Hong et al., 2024) employ dual encoders to capture features at varying levels of granularity. Other models like SPHINX (Lin et al., 2023) and Cambrian-1 (Tong et al., 2024a) have explored using up to four distinct encoders. Several works have focused on the architectural aspects of fusing information from multiple encoders. I-MoF (Tong et al., 2024b) uses separate projection layers for its two encoders, while Vary (Haoran et al., 2023) extends the vocabulary to manage inputs from different visual sources. Prismer (Liu et al., 2023a) utilizes an expert resampler for outputs from an ensemble of experts. CoMM (Jiang et al., 2024) investigated effective combinations, finding CLIP (Radford et al., 2021) and DINO (Oquab et al., 2023) to be potent, while noting that MAE (He et al., 2022) and DeiT (Touvron et al., 2021) performed less effectively as visual branches. More recently, CLIP-MOE (Zhang et al., 2024b) proposed a model-agnostic strategy for building CLIP with a mixture-of-experts approach. While these studies have significantly advanced the capabilities of MLLMs, their primary focus has been on achieving state-of-the-art performance or enabling new functionalities. The critical question of encoder redundancy, i.e., the extent to which additional encoders provide unique, non-overlapping information—has received less direct attention. Although works like Eagle (Shi et al., 2024) and Mousi (Fan et al., 2024) have reported diminishing returns, our work is the first to propose a formal framework and principled metrics (CUR, IG) to systematically quantify and diagnose this redundancy, thereby enabling a deeper understanding of the efficiency and necessity of each one.

### 2.3 VISUAL TOKEN SELECTION

To better exploit such complementary experts and avoid overwhelming the language model with redundant visual tokens, a growing line of work introduces explicit expert and token-selection mechanisms. MoVA adaptively routes and fuses task-specific vision experts via a coarse-to-fine mixture-of-vision-experts adaptor (Zong et al., 2024), while Mixture-of-Vision-Encoders style frameworks such as MOVE and Mixpert route each input to the most suitable encoder or expert branch instead of activating all experts uniformly (Skripkin et al., 2025; He et al., 2025). Orthogonal to expert routing, LEO-MINI performs conditional token reduction, consolidating a large pool of visual tokens into a compact, query-aware subset before feeding them to the LLM (Wang et al., 2025). While these works effectively mitigate inefficiency by either selecting experts or pruning tokens, they largely operate under the implicit assumption that integrated vision encoders inherently provide complementary value—a premise we challenge in this study. Through systematic encoder masking, we demonstrate pervasive redundancy (and even detrimental effects) among multi-encoder architectures, which existing selection strategies fail to quantify or address fundamentally.

# 3 METHODOLOGY

This section details our formal approach to investigating encoder redundancy. We first define the multi-encoder MLLM architecture (Section 3.1), then introduce our proposed metrics for quantifying redundancy (Section 3.2).

## 3.1 PROBLEM FORMULATION

We consider MLLMs based on the prevalent "ViT-adapter-LLM" architecture (Liu et al., 2024; Bai et al., 2025; Zhu et al., 2025). As illustrated in Figure 1, given an image $I$ and a text prompt $T$, the output response $Y$ of a multi-encoder MLLM with a set of $n$ vision encoders $\mathcal{E}_n = \{E_1, \ldots, E_n\}$ is generated as:

$$Y = f_{\mathcal{E}_n}(I, T) = \text{LLM}(\text{proj}(\text{fusion}(E_1(I), \cdots, E_n(I))), T), \quad (1)$$

where $\text{fusion}(\cdot)$ combines features from the different encoders (e.g., concatenation (Lu et al., 2024a; Tong et al., 2024b; Shi et al., 2024) or attention-based fusion (Li et al., 2024)) and $\text{proj}(\cdot)$ is an adapter that aligns visual features with the LLM's embedding space (Liu et al., 2024).

While multiple encoders can theoretically provide more comprehensive visual information, they also introduce noise, conflicting signals, or critically, redundant information. Such redundancy arises when encoders learn overlapping features or when some encoders supplies information already captured by others. We define encoder redundancy as a scenario where *including an encoder (or a subset of encoders) does not yield to a meaningful performance improvement, or even causes degradation.* Formally, encoder redundancy is observed if removing one or more encoders does not harm or even improve performance. This implies that the information from those encoder is either redundant or detrimental, making their computational cost and architectural complexity unjustified.

## 3.2 QUANTIFYING ENCODER CONTRIBUTION AND REDUNDANCY

To move beyond observations, we introduce two metrics to quantify the utility of each encoder within a multi-encoder system.

**Conditional Utilization Rate (CUR)**   The Conditional Utilization Rate (CUR) of an encoder $E_i$ measures its unique contribution relative to the full encoder set $\mathcal{E}_n$:

$$u(E_i) = \frac{\text{acc}(f_{\mathcal{E}_n}) - \text{acc}(f_{\mathcal{E}_n \setminus \{E_i\}})}{\text{acc}(f_{\mathcal{E}_n})}, \quad (2)$$

where $f_{\mathcal{E}_n \setminus \{E_i\}}$ denotes the MLLM with $E_i$ masked (e.g., replaced by a zero tensor), and $\text{acc}(\cdot)$ is the accuracy on benchmark evaluations (Appendix A). Since $\text{acc}(\cdot) \in [0, 1]$, $u(E_i) \in (-\infty, 1]$. A large positive $u(E_i)$ indicates a substantial unique contribution; values near zero imply redundancy; and negative values show that the encoder is detrimental, introducing conflicting or noisy features.

**Information Gap (IG)**   Building on CUR, we define the information gap $\Delta_{gap}$ for an encoder set $\mathcal{E}_n$ as:

$$\Delta_{gap}(\mathcal{E}_n) := \max_{i \in 1,\ldots,n} u(E_i) - \min_{j \in 1,\ldots,n} u(E_j). \quad (3)$$

The IG measures disparity in encoder contributions. A small $\Delta_{gap}$ suggests balanced utility across encoders, while a large $\Delta_{gap}$ highlights severe imbalance: some encoders are indispensable while others are redundant or harmful. Such imbalance indicates inefficiency in the architecture, as redundant encoders inflate computational cost without improving performance.

Together, CUR and IG provide a rigorous framework for quantifying encoder contributions and characterizing redundancy in multi-encoder MLLMs.

# 4 EXPERIMENTS

Our experiments are designed to: (1) empirically validate the existence of encoder redundancy scenarios across different MLLM architectures, and (2) apply our proposed CUR and IG metrics to

quantify the encoder redundancy. We first introduce the experiment setup in Section 4.1. Then, we quantitatively analyze the contribution of each vision encoder or combinations in Section 4.2. Finally, we analyze the key factors that contribute to this phenomenon in Section 4.3.

## 4.1 EXPERIMENTAL SETUP

**Baseline Models** Eagle (Shi et al., 2024) represents MLLMs designed with a larger ensemble of encoders (typically 4 or 5), including `CLIP` (Radford et al., 2021), `ConvNext` (Liu et al., 2022), `SAM` (Kirillov et al., 2023), `EVA-02` (Fang et al., 2024), and `Pix2Struct` (Lee et al., 2023). Eagle primarily uses channel concatenation for feature fusion. Cambrian-1 (Tong et al., 2024a) introduces a vision-centric approach with a novel fusion mechanism called Spatial Vision Aggregator (SVA). Rather than directly feeding all image tokens to the LLM, SVA uses cross-attention with learnable queries to integrate features from multiple encoders, including `CLIP` (Radford et al., 2021), `ConvNext` (Liu et al., 2022), `SigLIP` (Zhai et al., 2023) and `DINO` (Oquab et al., 2023), which offers a contrast to simpler concatenation methods and allows us to study if more sophisticated fusion can mitigate redundancy. These model architectures provide an excellent testbed for investigating redundancy in systems with many specialized encoders, allowing us to calculate CUR for each and assess the overall IG. We provide a more detailed introduction in Appendix B.

**Evaluation Benchmarks** To assess MLLM performance and analyze redundancy across diverse capabilities, we adopt the benchmark categorization proposed by Cambrian-1 (Tong et al., 2024a) which groups common benchmarks into four distinct categories based on a principal component analysis: (1) General; (2) Knowledge; (3) OCR & Chart; (4) Vision-Centric. Using these categories (detailed in Appendix A) allows for a nuanced understanding of how encoder redundancy might manifest differently depending on the task demands. All evaluations are performed using standardized protocols, primarily leveraging VLMEvalKit (Duan et al., 2025) for consistency.

## 4.2 EVIDENCE OF PERVASIVE ENCODER REDUNDANCY

In this section, we systematically probe the multi-encoder systems to demonstrate that significant redundancy is an inherent characteristic of current architectures.

**Performance Resilience to Encoder Ablation.** To evaluate redundancy, we consider all $2^n$ encoder combinations for a model with $n$ encoders. When an encoder is masked, we replace its output with a zero tensor of the same shape. Figure 2 shows the distribution of overall performance for several multi-encoder MLLMs with respect to the number of masked encoders. The results reveal a consistent trend: performance of multi-encoder MLLMs degrades gracefully rather than catastrophically as specific encoders are removed. For instance, the best-case performance of Eagle-X5 7B decreases by under $4\%$ when 3 specific encoders are masked; the optimal performance of Cambrian-1 8B is achieved with a subset of 3 vision encoders, which is $3.5\%$ higher than the full model. These findings validate that multi-encoder MLLMs can maintain most of their capabilities with only a subset of encoders, implying that additional encoders often yield diminishing returns and introduce computation inefficiency.

**Quantifying Specialization with CUR and IG.** We next employ CUR and IG to quantify the unique contribution of each encoder's unique contribution. Figure 3 presents the CUR across benchmark categories. The results indicate strong specialization for tasks such as OCR & Chart, where CUR values are extremely high. For example, in Eagle-X4 8B, `EVA-02` achieves a CUR of $92.89\%$ on OCR & Chart, and in the Cambrian-1 series, `ConvNext` contributes with a CUR above $70\%$. This demonstrates that specific encoders dominate OCR-related tasks. In contrast, for Knowledge and General categories, CUR values are much lower, suggesting that encoders provide more homogeneous semantic features and are largely interchangeable. Notably, some encoders exhibit negative CUR values. For instance, in Cambrian-1 8B, `SigLIP` attains a CUR of $-16\%$ on the Vision-Centric category, indicating that its inclusion is detrimental—likely introducing conflicting signals that the fusion mechanism cannot resolve. To assess disparity more directly, we analyze IG. Table 1 shows that models with more than two encoders tend to have larger IG, reflecting greater redundancy. This imbalance is most evident in OCR & Chart and Vision-Centric categories, consistent with CUR results: a single encoder typically dominates performance for a given task, while others

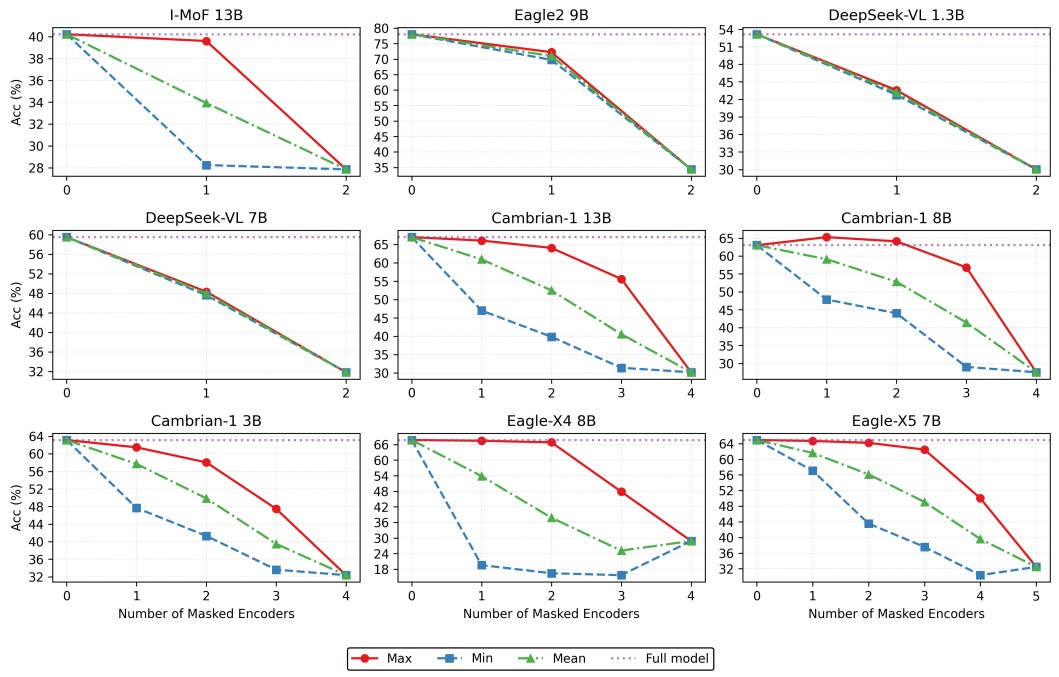

Figure 2: **Performance of multi-encoder MLLMs with different number of masked vision encoders**. Max, Min and Mean refer to the subset of ablated encoders with best, worst and average performance among all possible subsets respectively.

contribute minimally or not at all. As the result shows, MLLMs with more than 2 encoders tend to have larger IG, indicating that a greater number of encoders leads to increased redundancy. This imbalance is emphasized on OCR & Chart and Vision-Centric categories, which is consistent with CUR results, that is, a specific encoder dominates the contribution to a particular type of task, and this dominance is fixed, meaning that the model relies on this encoder while largely ignoring the others when performing on these tasks.

Table 1: **Information Gap of vision encoders on multi-encoder MLLMs**. A higher value indicates higher imbalance across different vision encoders.

| Model | $n$ | General | Knowledge | OCR & Chart | Vision-Centric | Overall |
|---|---|---|---|---|---|---|
| Eagle-X5 7B | 5 | 9.89% | 5.68% | 30.17% | 17.19% | 11.48% |
| Eagle-X4 8B Plus | 4 | 85.41% | 55.83% | 92.89% | 50.14% | 70.27% |
| Cambrian-1 3B | 4 | 7.30% | 8.09% | 66.47% | 8.45% | 22.42% |
| Cambrian-1 8B | 4 | 4.03% | 11.59% | 73.07% | 21.77% | 26.24% |
| Cambrian-1 13B | 4 | 9.62% | 14.87% | 76.22% | 12.28% | 27.82% |
| I-MoF 13B | 2 | 51.64% | 6.69% | 80.92% | 23.30% | 40.63% |
| Eagle2 9B | 2 | 10.50% | 0.23% | 28.03% | 8.79% | 2.24% |
| DeepSeek-VL 7B | 2 | 1.18% | 0.50% | 0.51% | 2.43% | 1.15% |

**Fewer Encoders, Comparable Performance.** Having established redundancy, we next examine whether comparable accuracy can be achieved with fewer encoders. We evaluate Eagle-X5 7B,

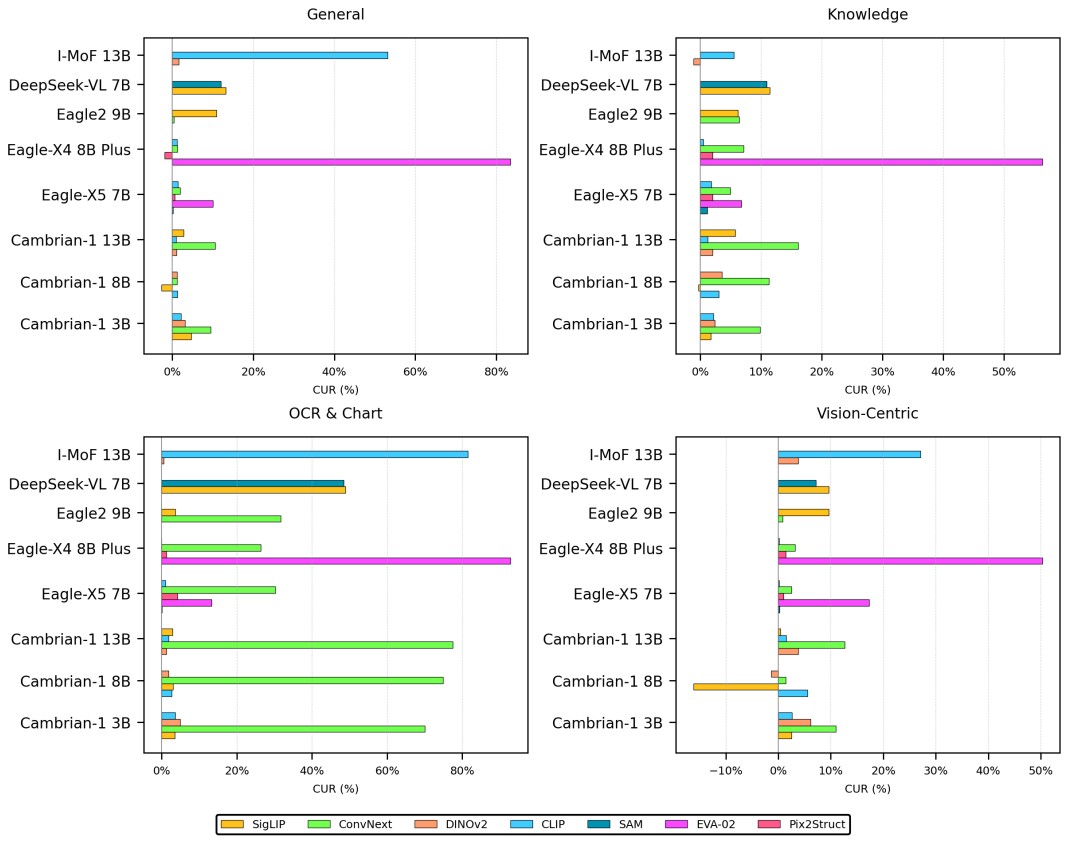

Figure 3: **CUR of different encoders across different category of benchmarks**. A higher CUR means a larger dependence on specific encoders.

Eagle-X4 8B Plus, and Cambrian-1 8B, in which `EVA-02` (Fang et al., 2024) and `ConvNext` (Liu et al., 2022) are the dominant contributors, respectively. Progressively masking encoders, we measure the resulting performance. As shown in Table 3, masking two encoders reduces Eagle model performance by only 1%, while the same operation increases Cambrian-1 8B performance by 1.7%. Across non-OCR tasks, single-encoder variants of the Eagle series retain at least 90% of the full model's performance with five encoders, showing that one strong encoder suffices for most capabilities. For OCR & Chart tasks, which demand fine-grained text and structural cues, adding `ConvNext` as a second branch substantially restores accuracy. From a resource perspective, additional encoders significantly inflate training costs: fine-tuning Eagle-X5 7B with five encoders requires approximately 94.57 A100 GPU hours, whereas removing three encoders reduces training time by 34.6% (See Appendix C for details). These results highlight a practical trade-off: employing one or two encoders recovers most accuracy while substantially lowering training time and computational cost.

Our CUR and IG metrics effectively quantify the severe imbalance in encoder contributions. Some encoders are highly specialized and indispensable for certain tasks like OCR & Chart (high CUR, high IG), while for other tasks like Knowledge-based tasks, encoders are largely interchangeable and redundant (low CUR, low IG). In summary, our analyses confirm that encoder redundancy is both pervasive and predictable, suggesting that careful encoder selection can preserve accuracy while substantially improving efficiency in multi-encoder MLLMs.

**Computation efficiency.** Beyond accuracy, removing redundant encoders brings tangible savings in both training and inference cost. Table 17 reports the wall-clock time (on 8 NVIDIA A100 GPUs) for pre-training and fine-tuning Eagle-X5 7B and its variants. Moving from the full five-encoder Eagle-X5 7B to its dual-encoder variant Eagle-X2 7B reduces the overall training time by

34% while retaining over 96% of the original performance. A similar trend holds at inference time. Table 18 reports per-sample latency on MME under different masking configurations. For Eagle-X5 7B, masking three encoders reduces end-to-end latency by 19.5% while maintaining more than 96% of its performance. Table 19 further decomposes the vision FLOPs of Eagle-X5 7B across encoders where `ConvNext` dominates the vision-side computation, and the full vision stack requires about 6.62 TFLOPs per image. For a typical yes/no QA with a 64-token prompt and a one-token answer, the vision processing accounts for roughly 31% of the total FLOPs. When we keep only the two most useful encoders identified by our CUR/IG analysis in Eagle-X2 7B, the vision-side FLOPs drop to 61.4% compared with Eagle-X5 7B, corresponding to a ∼12.1% reduction in total FLOPs and a 19.5% reduction in latency, with less than 4% average performance degradation. These measurements quantitatively demonstrate that eliminating redundant encoders yields substantial efficiency gains in both training and inference.

## 4.3 Analyzing on Encoder Redundancy

Previous experiments validate the existence of encoder redundancy. In this section, we investigate its underlying causes.

Table 2: **Attention score distribution analysis**. KL Divergence of attention score distributions between full model and the model with only one encoder activated. Lower value indicates higher Similarity.

| Model | CLIP | ConvNext | SAM | EVA-02 | Pix2Struct | SigLIP | DINOv2 |
|---|---|---|---|---|---|---|---|
| Eagle-X5 7B | 2.658 | 3.004 | 2.537 | **0.982** | 2.959 | - | - |
| Eagle-X4 8B Plus | 1.007 | ∞ | ∞ | **0.392** | - | - | - |
| Cambrian-1 8B | 0.102 | **0.080** | - | - | - | 0.095 | 0.128 |

**Attention Analysis.** While the IG values for the Eagle and Cambrian-1 series are relatively high, the outstanding CUR of `EVA-02` and `ConvNext` in these two series suggest that the outputs of these encoders dominate the visual representation respectively. To further investigate these disparities, we perform an attention-based analysis using the Eagle series and the Cambrian-1 8B model. Specifically, we evaluate each model on MME (Fu et al., 2024) with particular encoders selectively activated ($n = 1$), and extract the attention scores of visual tokens from the final layer of the LLM. Using the full model as a baseline, we then compute the Kullback–Leibler (KL) divergence between its attention distribution and those obtained when only a single encoder is activated. The results of this analysis are reported in Table 2. For Eagle series, the combination of high IG and the CUR dominance of `EVA-02` is reflected in the attention maps, which further confirm `EVA-02` as the primary contributor to the visual features. In contrast, for the Cambrian-1 series, `ConvNext` emerges as the most influential encoder. For Eagle-X4 8B Plus, the infinite KL for `ConvNext` and `SAM` indicates a support mismatch, where these single-encoder runs place zero mass on positions that the full model attends to, which is consistent with an `EVA-02`-dominated attention pattern. This analysis reveals a clear imbalance of encoder contributions during inference in multi-encoder MLLMs, with certain encoders contributing minimally, suggesting that some may be functionally redundant within the architecture. For more detailed analysis results, please refer to the Appendix C.

**The Role of Encoder Pre-training.** An encoder's pre-training objective and number of parameters largely governs the kind and quality of visual evidence it supplies to an MLLM. Firstly, the size of an encoder is irrelevant to its final contribution, `EVA-02`, an encoder with 304M parameters, dominates the performance in Eagle-X4 8B Plus compared to `Pix2Struct`, which has 1.2B parameters. Secondly, encoders pretrained with the same objective may ultimately lead to different levels of contribution. `ConvNext`, `CLIP` and `SigLIP` are all pretrained via a contrastive way, however, as shown in Figure 3, `ConvNext` achieves a higher CUR. Finally, different combinations of encoders may perform differently due to differences in model architecture or training data. For non-OCR tasks, `SigLIP` and `CLIP` show distinct interaction patterns within the Cambrian-1 series. These findings highlight a design trade-off: assembling diverse, specialized encoders can curb

Table 3: **Robustness of specific vision encoders with respect to masking operation**. Performance comparison of Eagle-X5 7B (`EVA-02`$_0$ + `ConvNext`$_1$ + `Pix2Struct`$_2$ + `CLIP`$_3$ + `SAM`$_4$), Eagle-X4 8B Plus (`EVA-02`$_0$ + `ConvNext`$_1$ + `Pix2Struct`$_2$ + `CLIP`$_3$), Cambrian-1 8B (`ConvNext`$_0$ + `DINOv2`$_1$ + `CLIP`$_2$ + `SigLIP`$_3$) and DeepSeek-VL 7B (`SAM`$_0$ + `SigLIP`$_1$) against masking operation. The subscript such as $_{0123}$ refers to retained encoder index.

| Model | $n$ | General | Knowledge | OCR & Chart | Vision-Centric | Overall |
|---|---|---|---|---|---|---|
| Eagle-X5 7B | 5 | 70.77 | 54.79 | 66.60 | 67.55 | 64.93 |
| −X4 $_{0123}$ | 4 | 70.64 ↓ 0 % | 54.19 ↓ 1 % | 66.55 ↓ 0 % | 67.39 ↓ 0 % | 64.69 ↓0.3% |
| −X3 $_{012}$ | 3 | 69.87 ↓ 1 % | 53.64 ↓ 2 % | 66.02 ↓ 1 % | 67.29 ↓ 0 % | 64.20 ↓1.1% |
| −X2 $_{01}$ | 2 | 69.04 ↓ 2 % | 52.77 ↓ 4 % | 62.04 ↓ 7 % | 66.05 ↓ 2 % | 62.48 ↓3.8% |
| −X1 $_{0}$ | 1 | 64.60 ↓ 9 % | 47.70 ↓13% | 10.68 ↓84% | 62.83 ↓ 7 % | 46.45 ↓ 28% |
| Eagle-X4 8B Plus | 4 | 66.49 | 61.88 | 71.92 | 70.62 | 67.73 |
| −X3 $_{012}$ | 3 | 65.68 ↓ 1 % | 61.57 ↓ 0 % | 71.97 ↑ 0 % | 70.50 ↓ 0 % | 67.43 ↓0.4% |
| −X2 $_{01}$ | 2 | 67.28 ↑ 1 % | 59.83 ↓ 2 % | 70.57 ↓ 1 % | 69.60 ↓ 0 % | 66.82 ↓1.1% |
| −X1 $_{0}$ | 1 | 64.22 ↓ 3 % | 51.21 ↓17% | 9.14 ↓83% | 66.68 ↓ 6 % | 47.81 ↓ 29% |
| Cambrian-1 8B | 4 | 67.47 | 57.88 | 70.08 | 56.65 | 63.02 |
| −X3 $_{012}$ | 3 | 69.29 ↑3 % | 58.05 ↑ 0 % | 67.97 ↓ 3 % | 65.81 ↑16% | 65.28 ↑3.6% |
| −X2 $_{03}$ | 2 | 68.64 ↑ 2 % | 58.09 ↑ 0 % | 66.41 ↓ 5 % | 63.24 ↑ 12 % | 64.09 ↑1.7% |
| −X1 $_{0}$ | 1 | 57.04 ↓15% | 53.72 ↓ 7 % | 60.57 ↓14% | 55.93 ↓ 1 % | 56.82 ↓9.8% |
| DeepSeek-VL 7B | 2 | 69.84 | 52.37 | 53.96 | 61.83 | 59.50 |
| −X1 $_{0}$ | 1 | 60.60 ↓ 13 % | 46.38 ↓ 11 % | 27.53 ↓ 49 % | 55.89 ↓10% | 47.60 ↓20% |
| −X1 $_{1}$ | 1 | 61.42 ↓ 12 % | 46.64 ↓ 11 % | 27.80 ↓ 48 % | 57.40 ↓ 7 % | 48.32 ↓19% |

redundancy on targeted skills, but risks under utilization and inefficiency on broad tasks; conversely, stacking multiple semantically similar encoders amplifies redundancy with limited aggregate gain.

**Number of Vision Encoders.** The number of encoders should also be considered when studying encoder redundancy. Both Eagle (Shi et al., 2024) and MouSi (Fan et al., 2024) performs ablation studies on number of encoders. According to MouSi, MLLMs with two encoders outperform those with a single encoder in most cases ($8/9$). However, when extended to three encoders, the winning case ratio drops to $4/6$. The results of the experiment in Table 3 shows a similar trend, that is, the dual-encoder architecture is a trade-off between performance and efficiency. When increasing the number of encoders, the performance improvement becomes marginal. When adopting only a single encoder, performance on specialized tasks such as OCR & Chart drops.

**LLM Capacity.** Our investigation into the impact of model scale on encoder redundancy, as shown in Table 1, reveals that while larger models achieve higher performance, they also exhibit more pronounced redundancy (See Table 15 for details). For Cambrian-1 series, masking a single encoder in the 13B model results in performance that can either remain as high as $98.6\%$ of the full model or drop to $70.2\%$, depending on which encoder is removed. This wide variance substantially larger than that observed in the 8B and 3B models, which indicates greater disparity in encoder contributions within the larger model. The ability of the 13B model to sustain near-peak performance even when its least important encoder is masked suggests a high degree of informational overlap. Consistently, the IG $\Delta_{gap}$ increases from $22.42\%$ to $27.82\%$ as the LLM size grows. These findings strongly suggest that encoder redundancy becomes more pronounced as LLMs scale up, rendering larger models both more robust to the loss of individual encoders and less efficient in their architectural design.

**Ablation Study on Masking Operation.** We selected zero-masking which replaces an encoder's output with a zero tensor for its simplicity in our main analysis. To validate this choice, we compare it with mean-masking, which instead uses the feature's mean value. While both methods perform similarly when masking few encoders, mean-masking is significantly more robust in the extreme single-encoder setting ($n = 1$), where it nearly matches the full model's performance (Table 4). The remarkable effectiveness of a simple mean value strongly highlights the redundancy of the other

Table 4: **Ablation study on masking operation**. Zero masking and mean masking replace specific encoders' output image features with the same-shaped zero tensors and their element-wise mean, respectively.

| Model | $n$ | MMBench | MMVP | ScienceQA | TextVQA |
|---|---|---|---|---|---|
| Eagle-X4 8B Plus | 4 | 71.39 | 71.00 | 80.16 | 66.29 |
| $-$X3 $_{012}$ (Zero) | 3 | 70.10 ↓2 % | 70.67 ↓0 % | 80.16 ↓0 % | 66.15 ↓0 % |
| $-$X2 $_{01}$ (Zero) | 2 | 70.62 ↓1 % | 70.67 ↓0 % | 79.64 ↓1 % | 65.91 ↓1 % |
| $-$X1 $_{0}$ (Zero) | 1 | 62.29 ↓13% | 66.00 ↓7 % | 72.06 ↓10% | 10.67 ↓84% |
| $-$X3 $_{012}$ (Mean) | 3 | 69.85 ↓2 % | 70.00 ↓1 % | 79.97 ↓0 % | 65.90 ↓1 % |
| $-$X2 $_{01}$ (Mean) | 2 | 69.85 ↓2 % | 69.00 ↓3 % | 79.59 ↓1 % | 65.86 ↓1 % |
| $-$X1 $_{0}$ (Mean) | 1 | 69.76 ↓2 % | 69.00 ↓3 % | 79.97 ↓0 % | 65.96 ↓0 % |

encoders. This confirms that our choice of the simpler zero-masking operation serves as an effective albeit more stringent method for quantifying encoder contribution.

**Discussion** The Platonic Representation Hypothesis Huh et al., 2024 posits that large models trained on similar data develop increasingly aligned representation spaces. Since multi-encoder MLLMs use vision encoders trained on overlapping, web-scale corpora, their feature spaces naturally converge. Our empirical finding—that adding more encoders yields diminishing returns and that encoders are largely interchangeable—is a direct manifestation of this representational convergence.

## 5 LIMITATION AND CONCLUSION

**Limitation** First, our performance evaluation focuses on standardized benchmark accuracy across four core multimodal task dimensions (General, Knowledge, OCR & Chart, Vision-Centric), without explicitly assessing robustness, out-of-distribution generalization, calibration, or fairness—dimensions requiring specialized datasets. While our CUR and IG metrics are metric-agnostic (extensible to these axes via task-specific scores), systematic validation of this extensibility demands dedicated protocols, which we leave for future work. Second, our analyses are limited to pre-trained, static multi-encoder MLLMs (fixed fusion, no dynamic routing). This choice leverages their status as mainstream foundational architectures and stable input-output mappings for reliable redundancy quantification. Dynamic routing, by contrast, causes variable encoder activation weights, destabilizing CUR calculations. We acknowledge our framework cannot disentangle intermediate redundancy from encoder-LLM alignment in dynamic models. Finally, a priori prediction of multi-encoder MLLM performance—e.g., scaling laws for encoder count—remains unresolved. Our CUR/IG metrics quantify redundancy post-hoc but not upfront encoder selection, constraining their use in initial architecture design. Notably, they still enable actionable post-training optimization. Establishing pre-training scaling relationships remains a key future direction.

**Conclusion** In this paper, we presented the first systematic, quantitative investigation into encoder redundancy in Multi-encoder MLLMs. We introduced the Conditional Utilization Rate (CUR) and Information Gap (IG) as principled metrics to quantify encoder contribution and utility disparity. Our comprehensive analysis, including combinatorial ablation and efficiency trade-off studies, revealed three core findings: (1) **Pervasive Redundancy**: Adding more encoders often yields severely diminishing returns, resulting in performance gains that are negligible compared to the substantial increase in required training resources (i.e., practical inefficiency); (2) **Dominance and Specialization**: Performance is often dominated by a single key encoder or a small complementary pair, leading to a large Information Gap (IG); (3) **Efficiency Breakthrough**: We demonstrated that dual-encoder variants can consistently recover over 90% of the full model's performance with significantly reduced computational cost. These results fundamentally challenge the "more is better" heuristic. Our work provides an actionable diagnostic framework (CUR/IG) to guide the design of future efficient, resource-optimized MLLMs.

## ACKNOWLEDGMENT

The research was supported by Shanghai Artificial Intelligence Laboratory.

## ETHICAL CONSIDERATIONS

**Ethical considerations**   Our study analyzes and optimizes multi-encoder MLLMs, which raises several ethical points. First, efficiency gains (e.g., comparable accuracy with fewer encoders) have positive environmental implications by reducing GPU hours and energy use, but they also lower deployment barriers; efficient models could be repurposed for privacy-sensitive applications (e.g., large-scale surveillance). We therefore discourage uses that violate privacy or local regulations and recommend task-appropriate safeguards (rate limiting, face/plate filtering, audit logs). Second, pruning or down-weighting encoders can introduce **distributional harm**: performance regressions may disproportionately affect specific scripts, chart styles, domains (e.g., OCR for low-resource languages), or accessibility use-cases. To mitigate this, we advocate reporting per-subset metrics (language, domain, layout) alongside overall scores, and avoiding model changes that degrade critical or safety-relevant tasks. Third, our evaluation relies on public benchmarks and LLM-based judging; these may encode social and cultural biases and can over- or under-estimate real-world reliability. We encourage complementary human evaluation and transparency about judge models and prompts. Fourth, we use only publicly licensed encoders and datasets and avoid training on personal or sensitive data; when working with document images, care should be taken to exclude PII and respect dataset licenses. Finally, revealing encoder-specific salience (via CUR/IG or attention analyses) could be exploited to craft targeted adversarial inputs; we release results at aggregate levels and recommend standard robustness testing before deployment. Overall, our efficiency recommendations should be paired with privacy, fairness, and robustness checks to ensure that benefits do not come at the expense of vulnerable users.

## CLAIM ON REPRODUCIBILITY

**Claim on Reproducibility**   We commit that all reported results are fully reproducible, where the code and models will be open-sourced upon acceptance.

## USE OF LARGE LANGUAGE MODELS

**Use of Large Language Models**   Large Language Models are mainly used for grammar check and polishing in this paper. The prompt used for polishing the paper is: "Please help me to check if there are grammar error, if possible, provide some advice on how to improve my academic paper."

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

## A BENCHMARK AND EVALUATION DETAILS

The evaluated benchmarks, summarized in Table 5, are classified into four categories:

- **OCR & Chart:** Emphasizes text extraction and reasoning from visual documents (e.g., ChartQA (Masry et al., 2022), OCRBench (Liu et al., 2023c), TextVQA (Singh et al., 2019), DocVQA (Mathew et al., 2021)).
- **Knowledge-based VQA:** Relies on world knowledge and complex reasoning integrated with visual perception (e.g., SQA (Lu et al., 2022), MMMU (Yue et al., 2024), Math-Vista (Lu et al., 2024b), AI2D (Hiippala et al., 2021)).
- **Vision-Centric Tasks:** Requires detailed visual understanding and focus on visual attributes (e.g., MMVP (Tong et al., 2024b), RealWorldQA (xAI, 2024), CV-Bench 2D/3D (Tong et al., 2024a)).
- **General VQA:** Assesses overall comprehensive visual understanding and reasoning (e.g., MME (Fu et al., 2024), MMBench (Liu et al., 2023b), SEED-Bench (Ge et al., 2023), GQA (Hudson & Manning, 2019)).

For consistency, all benchmarks are evaluated via VLMEvalKit (Duan et al., 2025), where `do_sample` is set to `False`; `num_beams` is set to 1 and `max_new_tokens` is set to 1024 by default. For MME benchmark, we only report its perception score to be consistent with (Tong et al., 2024a). Before averaging, we divide the MME score by 20 and OCRBench score by 10. We use Qwen-Plus as judge model when there is a need. Other configurations are kept default.

When compute `acc` of a model on these four categories, we first evaluate the performance on each benchmarks, then we compute the average over these benchmarks, for example, on General category, the accuracy is computed as

$$\mathrm{acc(model)} = \frac{\mathrm{acc(model)_{GQA} + acc(model)_{MMB} + acc(model)_{MME} + acc(model)_{SEED-I}}}{4}$$

(4)

Table 5: **Evaluation benchmark details**. 15 benchmarks are classified into 4 categories.

| Category | Benchmark | metric | remark |
|---|---|---|---|
| **General** | GQA (Hudson & Manning, 2019) | accuracy | |
| | MMB (Liu et al., 2023b) | accuracy | |
| | MME (Fu et al., 2024) | score | perception score divided by 20 |
| | SEED-I (Ge et al., 2023) | accuracy | |
| **Knowledge** | AI2D (Hiippala et al., 2021) | accuracy | |
| | MathVista (Lu et al., 2024b) | score | |
| | SQA-I (Lu et al., 2022) | accuracy | |
| | MMMU (Yue et al., 2024) | accuracy | |
| **OCR & Chart** | DocVQA (Mathew et al., 2021) | accuracy | |
| | ChartQA (Masry et al., 2022) | accuracy | |
| | OCRBench (Liu et al., 2023c) | score | divided by 10 |
| | TextVQA (Singh et al., 2019) | accuracy | |
| **Vision-Centric** | CV-Bench (Tong et al., 2024a) | accuracy | |
| | MMVP (Tong et al., 2024b) | accuracy | |
| | Real World QA (xAI, 2024) | accuracy | |

## B  MULTI-ENCODER MLLMS SUMMARIZATION

A bunch of related works combines the information extracted by multiple vision encoders, we summarize the MLLMs with multi encoders and their fusion strategies in Table 6. Most multi-encoders adopts `CLIP`, `SigLIP`, which are trained with contrastive objective to extract rich semantic information from visual inputs. To enhance the ability of processing low-level, fine-grained information, encoders such as `SAM`, `Pix2Struct` are utilized to enhance the capability of handling detection and OCR related tasks.

Table 6: **Fusion strategy and vision encoders used by MLLMs with multi encoders.** The adopted vision encoders and fusion strategies are presented.

| Model | Cambiran-1 (Tong et al., 2024a) | Mini-Gemini (Li et al., 2024) | Eagle-X4 (Shi et al., 2024) | Eagle-X5 (Shi et al., 2024) | Eagle2 (Li et al., 2025b) | Mousi (Fan et al., 2024) | deepseek-vl (Lu et al., 2024a) | Brave (Kar et al., 2024) | I-MoF (Tong et al., 2024b) | SPHINX (Lin et al., 2023) | MoME (Shen et al., 2024) | MoVA (Zong et al., 2024) | CoMM (Jiang et al., 2024) | Ferret-v2 (Zhang et al., 2024a) | LLaVA-HR (Luo et al., 2024) | LEO (Azadani et al., 2025) | LEO-mini (Wang et al., 2025) |
|---|---|---|---|---|---|---|---|---|---|---|---|---|---|---|---|---|---|
| Fusion | SVA | ATT | CC | CC | CC | MLP | CC | SA | I-MoF | CC | MoLE | MoV | SA | SA | MRA | MLP | COTR |
| CLIP | ✓ | ✓ | ✓ | ✓ | | | | ✓ | ✓ | ✓ | ✓ | ✓ | ✓ | ✓ | ✓ | | ✓ |
| SigLIP | ✓ | | | | ✓ | | ✓ | ✓ | | | | | | | | | |
| ConvNext | ✓ | ✓ | ✓ | ✓ | ✓ | | | | | | | ✓ | | | | | ✓ |
| DINOv2 | ✓ | | | | | | | ✓ | ✓ | ✓ | ✓ | ✓ | ✓ | ✓ | ✓ | | |
| Pix2Struct | | | ✓ | ✓ | | ✓ | | | | | | ✓ | ✓ | | | | |
| EVA-02 | | | ✓ | ✓ | | ✓ | | ✓ | | | | | | | | | ✓ |
| SAM | | | ✓ | | | ✓ | ✓ | | | | | | | | | ✓ | |
| ViT | | | | | | | | ✓ | | | | | | | | | |
| Co-DETR | | | | | | | | | | | | | | ✓ | | | |
| Deplot | | | | | | | | | | | | | | ✓ | | | |
| Vary | | | | | | | | | | | | | | ✓ | | | |
| BiomedCLIP | | | | | | | | | | | | | | ✓ | | | |
| Intern-ViT | | | | | | | | | | | | | | | | ✓ | |

## C  EXPERIMENT RESULT DETAILS

Table 7, Table 8, Table 9, Table 10, Table 11, Table 12, Table 13 and Table 14 shows the detailed performance of each combination of different multi-encoder MLLMs, as we can see, the best-case performance is attained with a few combinations of encoders, for example, in Eagle-X5 7B, with `ConvNext` and `EVA-02`, the performance achieves 96% of the baseline (with no encoders are masked), indicating that other encoders contributes little to final performance.

Table 15 gives a detailed performance of different LLM sizes against the number of masked encoders. As we can see, encoder redundancy does not reduce as we increasing the size of LLM, meaning that though larger LLM shows better performance, it still faces digesting redundant or conflicting visual signals.

Table 16 is an extended version of Table 2, which reveal the difference between two evaluation configuration under different metrics, results are consistent for most metrics, so we only report the KL divergence in this paper.

Table 17 shows the training time and corresponding performance of different variants of Eagle-X5 7B, as the table shows, a dual-encoder variant hits a good trade-off between efficiency and performance, where it achieves 94% performance of Eagle-X5 7B while reducing the total training time by 34.4%.

Table 18 shows the inference times with respect to different masking operations, as the result shows, by masking redundant encoders, the performance is largely retained and the efficiency is improved. Table 19 shows computation cost of each vision encoder of Eagle-X5 7B, the FLOPs of vision encoders is FLOPs_encoders $\approx$ 6.6174 TFLOPs, the FLOPs of projection layer is FLOPs_proj $\approx$ 0.07 TFLOPs and the FLOPs of LLM is FLOPs_LLM $\approx$ 21.3 TFLOPs, for a full yes-or-no QA task (64 prompt tokens + 1 answer token, excluding special tokens), visual encoder processing accounts for approximately 30.6% of the total compute. Retaining only 2 encoders reduces total compute by $(1-14.1\%-47.3\%)*30.6\% = 11.8\%$ and inference latency by 19.5%, while limiting performance degradation to no more than 4%.

Table 20 detailed the performance of trained ones compared with the full model. Results show that the masked one performs slightly better than the trained one, revealing that incorporating more encoders may help on gaining more knowledge, however the improvement is marginal and at a cost of consuming more training and inference resources.

Table 7: **Benchmark details for Eagle-X5 7B**. Encoders: `1.CLIP`; `2.ConvNext`; `3.SAM`; `4.EVA-02`; 5. `Pix2Struct`. ✓means the corresponding encoder is masked. Best scores within each subset of encoders have been bolded.

| #Masked | Encoders | Performance | | | |
|---|---|---|---|---|---|
| | | **General** | **Knowledge** | **OCR & Chart** | **Vision-Centric** |
| 0 | × × × × × | 70.77 | 54.79 | 66.60 | 67.54 |
| 1 | × × ✓ × × | **70.64** | **54.18** | **66.54** | 67.38 |
| | × × ×✓× | 63.63 | 51.07 | 57.71 | 55.83 |
| | × × × × ✓ | 70.36 | 53.67 | 63.80 | 66.84 |
| | ×✓ × × × | 69.40 | 52.07 | 46.44 | 65.82 |
| | ✓ × × × × | 69.79 | 53.79 | 65.92 | **67.44** |
| 2 | × × ✓ × ✓ | **70.21** | **53.78** | 63.60 | 66.51 |
| | × × ✓✓× | 63.07 | 50.73 | 57.46 | 54.82 |
| | × × ×✓✓ | 61.65 | 49.62 | 44.58 | 52.92 |
| | ✓ × × × ✓ | 68.97 | 52.40 | 62.05 | 65.97 |
| | ✓✓ × × × | 65.69 | 51.35 | 40.81 | 64.53 |
| | ✓ × ×✓× | 59.24 | 51.14 | 55.92 | 52.68 |
| | ×✓✓ × × | 69.39 | 52.51 | 46.22 | 65.59 |
| | ×✓ × ✓× | 53.88 | 47.76 | 24.97 | 47.56 |
| | ✓ × ✓ × × | 69.86 | 53.64 | **66.01** | **67.29** |
| | ×✓ × ×✓ | 68.22 | 50.62 | 16.53 | 65.29 |
| 3 | ✓ × ✓ × ✓ | **69.04** | **52.77** | **62.04** | **66.05** |
| | ✓✓ × ×✓ | 64.97 | 47.96 | 10.84 | 62.91 |
| | ✓✓ × ✓× | 33.39 | 45.85 | 27.33 | 43.71 |
| | ✓✓✓ × × | 65.32 | 51.16 | 40.26 | 64.06 |
| | ×✓✓✓× | 53.07 | 47.50 | 24.73 | 46.84 |
| | × × ✓✓✓ | 61.13 | 49.26 | 44.19 | 51.84 |
| | ✓ × ×✓✓ | 58.08 | 48.88 | 44.20 | 50.76 |
| | ✓ × ✓✓× | 58.45 | 50.63 | 55.57 | 51.92 |
| | ×✓ × ✓✓ | 53.27 | 47.25 | 10.07 | 47.59 |
| | ×✓✓ × ✓ | 68.17 | 50.63 | 16.25 | 65.39 |
| 4 | ✓ × ✓✓✓ | 56.91 | **48.95** | **44.18** | 49.99 |
| | ×✓✓✓✓ | 51.94 | 47.47 | 10.13 | 46.87 |
| | ✓✓✓✓× | 28.07 | 34.78 | 15.30 | 43.21 |
| | ✓✓ × ✓✓ | 31.03 | 45.91 | 7.653 | 43.64 |
| | ✓✓✓ × ✓ | **64.60** | 47.70 | 10.68 | **62.83** |
| 5 | ✓✓✓✓✓ | 31.93 | 43.69 | 7.521 | 46.70 |

Table 8: **Benchmark details for Eagle-X4 8B Plus**. Encoders: 1.`CLIP`; 2.`ConvNext`; 3.`PIX2STRUCT`; 4.`EVA-02`. ✓means the corresponding encoder is masked. Best scores within each subset of encoders have been bolded.

| #Masked | Encoders | Performance | | | |
|---|---|---|---|---|---|
| | | **General** | **Knowledge** | **OCR & Chart** | **Vision-Centric** |
| 0 | ×××× | 66.48 | 61.88 | 71.92 | 70.62 |
| 1 | ✓××× | 65.68 | **61.57** | **71.97** | **70.50** |
| | ×✓×× | 65.60 | 57.46 | 52.89 | 68.38 |
| | ×××✓ | 10.99 | 27.03 | 5.165 | 35.09 |
| | ××✓× | **67.77** | 0.64 | 70.98 | 69.58 |
| 2 | ×✓×✓ | 7.813 | 23.93 | 1.180 | 32.85 |
| | ✓✓×× | 62.86 | 56.00 | 52.10 | 68.01 |
| | ✓××✓ | 6.905 | 33.60 | 0.175 | 33.78 |
| | ✓×✓× | **67.28** | **59.83** | **70.57** | **69.60** |
| | ×✓✓× | 65.85 | 51.60 | 10.17 | 66.93 |
| | ××✓✓ | 6.501 | 24.27 | 0.980 | 34.47 |
| 3 | ✓✓✓× | **64.22** | **51.21** | **9.141** | **66.68** |
| | ✓✓×✓ | 6.840 | 26.74 | 1.000 | 28.26 |
| | ×✓✓✓ | 6.604 | 26.67 | 0.125 | 38.66 |
| | ✓×✓✓ | 6.860 | 28.16 | 1.970 | 39.23 |
| 4 | ✓✓✓✓ | 25.60 | 44.54 | 5.316 | 40.00 |

Table 9: **Benchmark details for Cambrian-1 3B**. Encoders: 1.`CLIP`; 2.`SigLIP`; 3.`DINO`; 4.`ConvNext`; ✓means the corresponding encoder is masked. Best scores within each subset of encoders have been bolded.

| #Masked | Encoders | Performance | | | |
|---|---|---|---|---|---|
| | | **General** | **Knowledge** | **OCR & Chart** | **Vision-Centric** |
| 0 | ×××× | 67.81 | 59.58 | 60.60 | 64.57 |
| 1 | ✓××× | 64.65 | **58.54** | **58.44** | **62.93** |
| | ×✓×× | **66.34** | 58.31 | 58.36 | 62.88 |
| | ×××✓ | 61.39 | 53.72 | 18.16 | 57.48 |
| | ××✓× | 65.70 | 58.15 | 57.61 | 60.57 |
| 2 | ×✓×✓ | 51.45 | 51.22 | 10.52 | 52.08 |
| | ✓✓×× | 55.02 | 55.57 | 53.35 | 57.77 |
| | ✓××✓ | 52.14 | 51.14 | 12.01 | 53.87 |
| | ✓×✓× | 62.29 | 56.67 | **54.92** | 57.21 |
| | ×✓✓× | **63.28** | **56.88** | 53.80 | **58.24** |
| | ××✓✓ | 55.54 | 52.18 | 16.83 | 52.65 |
| 3 | ✓✓✓× | 41.03 | **52.84** | **47.01** | **49.01** |
| | ✓✓×✓ | 31.48 | 47.62 | 6.896 | 48.52 |
| | ×✓✓✓ | **44.84** | 50.34 | 9.510 | 48.96 |
| | ✓×✓✓ | 44.35 | 49.97 | 11.14 | 48.27 |
| 4 | ✓✓✓✓ | 29.26 | 47.81 | 6.586 | 45.88 |

Table 10: **Benchmark details for Cambrian-1 8B**. Encoders: 1.`CLIP`; 2.`SigLIP`; 3.`DINO`; 4.`ConvNext`; ✓means the corresponding encoder is masked. Best scores within each subset of encoders have been bolded.

| #Masked | Encoders | Performance | | | |
|---------|----------|---------|-----------|-------------|----------------|
|         |          | **General** | **Knowledge** | **OCR & Chart** | **Vision-Centric** |
| 0 | × × × × | 67.47 | 57.87 | 70.08 | 56.65 |
| 1 | ✓ × × × | 66.57 | 56.13 | 68.19 | 53.47 |
|   | ×✓ × × | **69.29** | **58.05** | 67.96 | **65.80** |
|   | × × ×✓ | 66.69 | 51.34 | 17.53 | 55.84 |
|   | × × ✓× | 66.68 | 55.83 | **68.74** | 57.43 |
| 2 | ×✓ × ✓ | 60.05 | 48.98 | 10.70 | 57.53 |
|   | ✓✓ × × | 59.77 | 52.64 | 63.91 | 50.31 |
|   | ✓ × ×✓ | 59.35 | 50.89 | 10.62 | 55.16 |
|   | ✓ × ✓× | **68.63** | **58.08** | 66.40 | **63.24** |
|   | ×✓✓× | 64.91 | 53.92 | **66.75** | 55.61 |
|   | × × ✓✓ | 64.28 | 51.61 | 16.71 | 56.95 |
| 3 | ✓✓✓× | **57.04** | **53.72** | **60.57** | **55.93** |
|   | ✓✓ × ✓ | 26.52 | 45.81 | 5.486 | 38.30 |
|   | ×✓✓✓ | 51.75 | 47.78 | 10.14 | 49.39 |
|   | ✓ × ✓✓ | 52.58 | 50.10 | 10.13 | 47.38 |
| 4 | ✓✓✓✓ | 23.33 | 45.06 | 5.914 | 35.83 |

Table 11: **Benchmark details for Cambrian-1 13B**. Encoders: 1.`CLIP`; 2.`SigLIP`; 3.`DINO`; 4.`ConvNext`; ✓means the corresponding encoder is masked. Best scores within each subset of encoders have been bolded.

| #Masked | Encoders | Performance | | | |
|---------|----------|---------|-----------|-------------|----------------|
|         |          | **General** | **Knowledge** | **OCR & Chart** | **Vision-Centric** |
| 0 | × × × × | 71.96 | 60.76 | 69.99 | 65.21 |
| 1 | ✓ × × × | 69.91 | 57.28 | 67.93 | **64.96** |
|   | ×✓ × × | **71.24** | **60.03** | 68.68 | 64.21 |
|   | × × ×✓ | 64.32 | 50.99 | 15.72 | 56.95 |
|   | × × ✓× | 71.20 | 59.54 | **69.07** | 62.70 |
| 2 | ×✓ × ✓ | 56.75 | 49.39 | 9.838 | 51.76 |
|   | ✓✓ × × | 63.02 | 56.45 | 63.47 | 61.86 |
|   | ✓ × ×✓ | 51.99 | 47.60 | 9.222 | 50.61 |
|   | ✓ × ✓× | 68.71 | 56.44 | 67.19 | **62.89** |
|   | ×✓✓× | **69.18** | **57.58** | **67.34** | 62.07 |
|   | × × ✓✓ | 58.89 | 50.10 | 14.83 | 52.27 |
| 3 | ✓✓✓× | **54.16** | **52.77** | **60.18** | **55.21** |
|   | ✓✓ × ✓ | 30.67 | 46.50 | 5.781 | 42.68 |
|   | ×✓✓✓ | 50.62 | 48.02 | 9.286 | 48.20 |
|   | ✓ × ✓✓ | 43.57 | 47.64 | 8.686 | 45.13 |
| 4 | ✓✓✓✓ | 27.92 | 45.96 | 5.806 | 41.06 |

Table 12: **Benchmark details for DeepSeek-VL 7B**. Encoders: 1.SAM; 2.SigLIP; ✓means the corresponding encoder is masked. Best scores within each subset of encoders have been bolded.

| #Masked | Encoders | Performance | | | |
|---|---|---|---|---|---|
| | | **General** | **Knowledge** | **OCR & Chart** | **Vision-Centric** |
| 0 | ×× | 69.84 | 52.37 | 53.96 | 61.83 |
| 1 | ✓× | **61.42** | **46.64** | **27.80** | **57.40** |
| | ×✓ | 60.60 | 46.38 | 27.53 | 55.89 |
| 2 | ✓✓ | 31.09 | 42.71 | 6.771 | 46.69 |

Table 13: **Benchmark details for Eagle2 9B**. Encoders: 1.SAM; 2.SigLIP; ✓means the corresponding encoder is masked. Best scores within each subset of encoders have been bolded.

| #Masked | Encoders | Performance | | | |
|---|---|---|---|---|---|
| | | **General** | **Knowledge** | **OCR & Chart** | **Vision-Centric** |
| 0 | ×× | 75.61 | 75.02 | 86.53 | 74.95 |
| 1 | ✓× | 67.36 | **70.39** | **83.37** | 67.75 |
| | ×✓ | **75.30** | 70.22 | 59.12 | **74.34** |
| 2 | ✓✓ | 30.99 | 51.57 | 6.542 | 48.31 |

Table 14: **Benchmark details for I-MoF 13B**. Encoders: 1.CLIP; 2.DINOv2; ✓means the corresponding encoder is masked. Best scores within each subset of encoders have been bolded.

| #Masked | Encoders | Performance | | | |
|---|---|---|---|---|---|
| | | **General** | **Knowledge** | **OCR & Chart** | **Vision-Centric** |
| 0 | ×× | 47.89 | 48.99 | 4.325 | 59.70 |
| 1 | ✓× | 22.41 | 46.27 | 0.800 | 43.53 |
| | ×✓ | **47.14** | **49.55** | **4.300** | **57.44** |
| 2 | ✓✓ | 24.17 | 45.47 | 0.725 | 49.26 |

Table 15: **Impact of the size of LLM on encoder redundancy**. The min, max and average performance of Cambrian-1 3B, 8B, 13B with different number of masked encoders are reported.

| LLM | #masked | 0 | 1 | 2 | 3 | 4 |
|---|---|---|---|---|---|---|
| 3B | max | 63.14 | 61.47 | 58.04 | 47.47 | 32.38 |
| | min | 63.14 | 47.68 | 41.31 | 33.62 | 32.38 |
| | avg | 63.14 | 57.70 | 49.86 | 39.48 | 32.38 |
| 8B | max | 63.02 | 65.28 | 64.09 | 56.81 | 27.53 |
| | min | 63.02 | 47.85 | 44.00 | 29.03 | 27.53 |
| | avg | 63.02 | 59.10 | 52.79 | 41.41 | 27.53 |
| 13B | max | 66.98 | 66.04 | 64.04 | 55.58 | 30.18 |
| | min | 66.98 | 46.99 | 39.85 | 31.40 | 30.18 |
| | avg | 66.98 | 60.92 | 52.47 | 40.56 | 30.18 |

Table 16: **Attention Analysis on different encoders**. We recore the value of last-layer attention scores related to visual tokens during a full MME evaluation, and we imply Pearson Correlation coefficient (PC), Spearman's Correlation (SC), Mean Squared Error (MSE), Mean Absolute Error (MAE), Jensen–Shannon divergence (JS) and Kullback-Leibler divergence (KL) for visual attention analysis. Higher PC, SC and lower MSE, MAE, JS, KL values indicates higher similarity. Eagle-X5 7B, Eagle-X4 8B Plus and Cambrian-1 7B are used for analysis.

| Model | Encoder | PC ↑ | SC ↑ | MSE ↓ | MAE ↓ | JS ↓ | KL ↓ |
|---|---|---|---|---|---|---|---|
| Eagle-X5 7B | CLIP | .0277 | .3878 | .00007 | .0013 | .5729 | 2.658 |
| | ConvNext | .0279 | .3726 | .00007 | .0013 | .5700 | 3.004 |
| | SAM | .0578 | .3712 | .00007 | .0012 | .5496 | 2.537 |
| | EVA | **.7446** | **.7432** | **.00003** | **.0006** | **.3619** | **0.982** |
| | Pix2Struct | .1934 | .3512 | .00015 | .0013 | .6082 | 2.959 |
| Eagle-X4 8B Plus | CLIP | .2742 | .4180 | .00003 | .0010 | .4353 | 1.007 |
| | ConvNext | .5854 | .2674 | .00002 | .0010 | .4146 | inf |
| | Pix2Struct | **.8281** | .4377 | .00001 | .0008 | .3553 | inf |
| | EVA | .7695 | **.7545** | **.00001** | **.0006** | **.2760** | **.3921** |
| Cambrian-1 8B | SigLIP | .9419 | .5468 | .00005 | .0009 | .1927 | .0951 |
| | CLIP | .9434 | .5453 | .00004 | .0009 | .1906 | .1017 |
| | DINOv2 | .9390 | .4956 | .00004 | .0009 | .1958 | .1279 |
| | ConvNext | **.9678** | **.6157** | **.00003** | **.0006** | **.1436** | **.0804** |

Table 17: **Training time with specific vision encoders.**. Training time of Eagle-X5 7B ($EVA-02_0$ + $ConvNext_1$ + $Pix2Struct_2$ + $CLIP_3$ + $SAM_4$) in hours, including pre-train stage and fine-tune stage. All experiments are conducted on a single-node with 8 NVIDIA A100 GPUs. The subscript such as $_{0123}$ refers to retained encoder index.

| Model | $n$ | Pre-train | Fine-tune | Overall | Performance |
|---|---|---|---|---|---|
| Eagle-X5 7B | 5 | 12.12 | 94.57 | 104.72 | 64.9 |
| $-X4_{0123}$ | 4 | 10.90 ↓10.0% | 81.92 ↓13.4% | 92.82 ↓11.4% | 62.06 ↓4.38% |
| $-X2_{01}$ | 2 | 6.888 ↓42.8% | 61.83 ↓34.6% | 68.71 ↓34.4% | 60.95 ↓6.09% |
| $-X1_0$ | 1 | 6.042 ↓49.8% | 58.20 ↓38.5% | 64.24 ↓38.7% | 51.57 ↓20.5% |

Table 18: **Inference time of specific vision encoders with respect to masking operation**. Inference time comparison of Eagle-X5 7B (EVA-02$_0$ + ConvNext$_1$ + Pix2Struct$_2$ + CLIP$_3$ + SAM$_4$) and Eagle-X4 8B Plus (EVA-02$_0$ + ConvNext$_1$ + Pix2Struct$_2$ + CLIP$_3$), evaluated by calculating the average inference time for MME benchmark evaluation with 2374 samples in ms, where the max number of generated tokens has ben set to 1. The subscript such as $_{0123}$ refers to retained encoder index.

| Model | $n$ | Time | Performance |
|---|---|---|---|
| Eagle-X5 7B | 5 | 749.656 | 64.93 |
| −X4 $_{0123}$ | 4 | 730.761 ↓2.52% | 64.69 ↓0.37% |
| −X3 $_{012}$ | 3 | 643.989 ↓14.1% | 64.20 ↓1.12% |
| −X2 $_{01}$ | 2 | 603.153 ↓19.5% | 62.48 ↓3.78% |
| −X1 $_0$ | 1 | 581.260 ↓22.5% | 46.45 ↓28.5% |
| Eagle-X4 8B Plus | 4 | 681.679 | 67.73 |
| −X3 $_{012}$ | 3 | 665.946 ↓2.31% | 67.43 ↓0.44% |
| −X2 $_{01}$ | 2 | 600.126 ↓12.0% | 66.82 ↓1.34% |
| −X1 $_0$ | 1 | 549.621 ↓19.4% | 47.81 ↓29.4% |

Table 19: **Per-encoder vision FLOPs for Eagle-X5 7B**. Including input-image resolution, vision FLOPs and contribution to Eagle-X5 7B vision computing.

| Encoder | Input Resolution | Vision FLOPs (G) | Contribution |
|---|---|---|---|
| CLIP-VIT-L/14 | 448 | 214.0 | 3.23% |
| ConvNext-Large | 1024 | 3133. | 47.3% |
| EVA-02-L | 1024 | 933.7 | 14.1% |
| Pix2Struct-1024 | 1024 | 1026. | 15.5% |
| SAM-ViT-1024 | 1024 | 1311. | 19.8% |

Table 20: **Performance for masked Eagle models with 1, 2, 4 and 5 encoders correspond to trained ones**. Performance subsets based on Eagle-X5 7B (EVA-02$_0$ + ConvNext$_1$ + Pix2Struct$_2$ + CLIP$_3$ + SAM$_4$), where all training strategies strictly follows the official instruction of training Eagle-X5 7B. The subscript such as $_{0123}$ refers to retained encoder index.

| Model | $n$ | General | Knowledge | OCR & Chart | Vision-Centric | Overall |
|---|---|---|---|---|---|---|
| X5 (Masked) $_{0123}$ | 4 | 70.64 | 54.19 | 66.55 | 67.39 | 64.69 |
| X5 (Trained) $_{0123}$ | 4 | 69.72 ↓1.30% | 49.25 ↓9.11% | 63.44 ↓4.67% | 65.85 ↓2.29% | 62.06 ↓4.07% |
| X5 (Masked) $_{01}$ | 2 | 69.04 | 52.77 | 62.04 | 66.05 | 62.48 |
| X5 (Trained) $_{01}$ | 2 | 67.22 ↓2.64% | 52.37 ↓0.76% | 60.17 ↓3.01% | 64.05 ↓3.03% | 60.95 ↓2.45% |

# D  CASE STUDY

We provide case study in this section. Results in Fig D and Fig D show that once a specific encoder is masked, the answer generated by the model changes dramatically, which reveals the fact that multi-encoder MLLM depends on specific encoder to complete specific tasks.

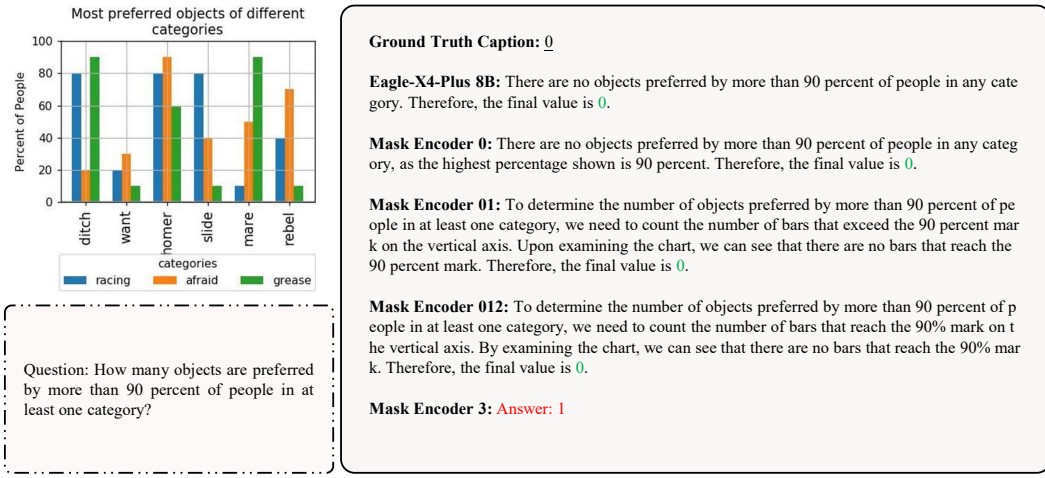

Figure 4: A show case of how Eagle-X4 8B Plus would behave with/without specific encoders masked.

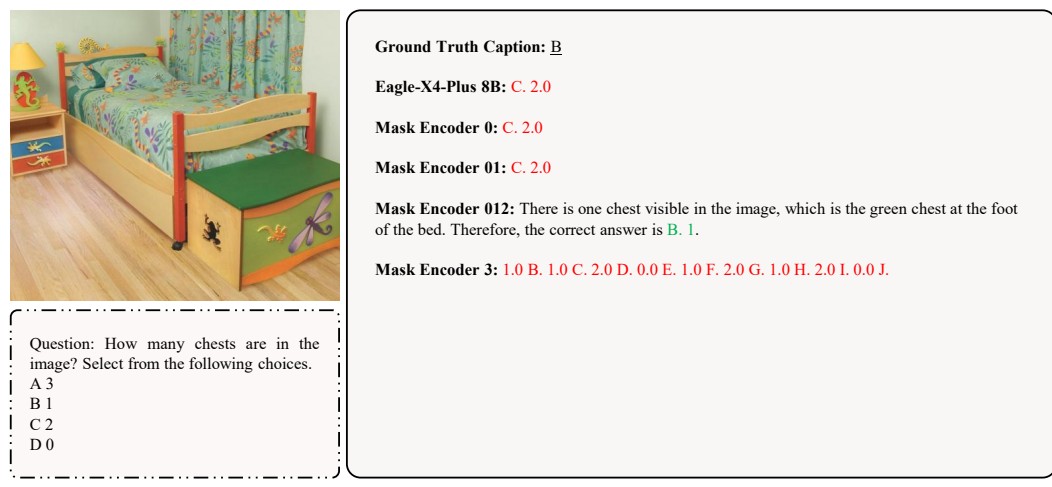

Figure 5: A show case of how Eagle-X4 8B Plus would behave with/without specific encoders masked.

