# OpenReview forum: "Investigating Redundancy in Multimodal Large Language Models with Multiple Vision Encoders"
_ICLR.cc/2026/Conference — ICLR 2026 Poster_

### Official Review · Reviewer_VGcs · 2025-10-24

**Soundness:** 3
**Presentation:** 3
**Contribution:** 2
**Rating:** 6
**Confidence:** 3

**Summary:**

This paper presents an investigation into the phenomenon of encoder redundancy in Multi-encoder Multimodal Large Language Models (MLLMs). The authors challenge the prevailing assumption that integrating more vision encoders, pre-trained with diverse objectives, necessarily leads to better performance. Through extensive ablation studies on models like Eagle and Cambrian-1, they demonstrate that performance often degrades gracefully—and sometimes even improves—when one or more encoders are masked. To quantify this redundancy, they introduce two metrics: the Conditional Utilization Rate (CUR), which measures an encoder's marginal contribution given the presence of others, and the Information Gap (IG), which captures the disparity in utility across the encoder set.

**Strengths:**

1. The paper addresses a critical, underexplored, and timely issue in MLLM design. As models grow more complex, understanding and mitigating redundancy is essential for improving efficiency and performance.
2. The proposed metrics, CUR and IG, are intuitive, easy to compute, and provide a rigorous, quantitative framework for analyzing encoder contributions. They effectively capture both individual utility and overall system imbalance.
3. The empirical evaluation is thorough. The authors test their hypothesis across multiple state-of-the-art model families (Eagle, Cambrian-1), various model sizes (3B to 13B), and a comprehensive set of benchmarks categorized by task type. The results are robust and clearly support the claims.

**Weaknesses:**

1. The study is conducted on pre-trained, static models. It does not explore whether redundancy emerges during training or if dynamic, input-dependent routing of encoders (a potential solution) could better leverage the available encoders.
2. **Narrow Scope of "Performance"**: Performance is primarily measured as accuracy on standardized benchmarks. Other critical dimensions like robustness, out-of-distribution generalization, calibration, and fairness are not considered. A redundant encoder might contribute to robustness even if it hurts average accuracy.
3. **Higher-Order Interactions**: The authors correctly identify this as a limitation. CUR only measures the effect of ablating a single encoder. It cannot detect complementary pairs or higher-order synergies where two seemingly redundant encoders (low individual CUR) are both essential for performance (e.g., if removed together, performance collapses). A metric capturing these interactions would be a valuable extension.
4. **Limited Exploration of Fusion Mechanisms**: While the paper notes that more sophisticated fusion (like Cambrian-1's SVA) does not fully mitigate redundancy, a deeper analysis is lacking. It remains unclear how different fusion strategies (e.g., MoE, cross-attention) might amplify or suppress redundancy, which is a crucial aspect of multi-encoder architecture design.

**Questions:**

1. How would you suggest extending the CUR framework to quantify the complementary or synergistic effects between pairs or groups of encoders, to address the limitation you noted?
2. What is the interplay between fusion mechanism complexity and redundancy? Could a better fusion strategy (e.g., a gating network) dynamically suppress redundant or conflicting encoders on a per-input basis, thereby making a larger ensemble more justifiable?

---

> ### Author Response · Authors · 2025-11-18
> **Part1**
>
> > Weaknesses 1: _The study is conducted on pre-trained, static models. It does not explore whether redundancy emerges during training or if dynamic, input-dependent routing of encoders (a potential solution) could better leverage the available encoders._
>
> Thank you for this valuable comment. Our current study is indeed conducted on pre-trained, fixed multi-encoder MLLMs, and our primary goal is to provide a diagnostic perspective on encoder redundancy at convergence. Extending CUR and IG to the training dynamics is non-trivial: early in training, the dominant factor is how well each encoder is aligned with the language model, rather than how redundant their final representations are. Because optimization is stochastic and the model is continually rebalancing encoder–LLM alignment, the level of “redundancy” at intermediate checkpoints is difficult to interpret and quantify in a stable way without very dense checkpointing, multiple training runs, and carefully controlled experimental setups. We agree this is an interesting direction, but it falls beyond the scope and computational budget of the present work; we now explicitly mention this as a promising avenue for future research, where one could study how redundancy evolves as encoders and the LLM gradually align during training.
>
> Regarding dynamic, input-dependent routing of encoders (like MoVA), we fully agree that such mechanisms are a natural way to use multiple encoders more efficiently. However, implementing and fairly evaluating such routing would require designing new architectures and re-training large multi-encoder models end-to-end, which is orthogonal to our objective here. In this paper, we deliberately focus on analyzing existing pre-trained multi-encoder MLLMs under fixed configurations and showing how CUR/IG can reveal when additional encoders become redundant or even harmful. We have clarified this positioning in the revised manuscript and explicitly highlight that our metrics are intended as building blocks that future work can leverage to guide dynamic encoder selection or routing schemes.
>
> ---
>
> > Weaknesses 2: _Narrow Scope of "Performance": Performance is primarily measured as accuracy on standardized benchmarks. Other critical dimensions like robustness, out-of-distribution generalization, calibration, and fairness are not considered. A redundant encoder might contribute to robustness even if it hurts average accuracy._
>
> We appreciate the reviewer’s point that “performance” is multi-dimensional and goes beyond average accuracy. In this work, we focus on standardized benchmark accuracy because:
>
> 1. we evaluate on 16 widely adopted benchmarks spanning diverse domains and formats (yes/no, multiple-choice, captioning, etc.), and aggregate them into four semantic dimensions (General, Knowledge, OCR & Chart, and Vision-Centric)
>
> 2. these are exactly the metrics used by prior multi-encoder MLLMs such as Eagle and Cambrian-1. Aligning with this evaluation protocol allows us to make a fair and direct comparison and to clearly isolate the effect of encoder redundancy under the same conditions as existing work.
>
> We fully agree that other aspects—robustness, out-of-distribution generalization, calibration, and fairness—are also important, and we do not claim that our analysis exhaustively covers these dimensions. Our CUR and IG metrics are, in principle, agnostic to the underlying metric and could be applied to robustness or calibration scores if suitable evaluations are available. However, systematically studying how redundant encoders might trade off average accuracy against robustness or fairness would require additional specialized benchmarks and stress-test setups, which we view as valuable but beyond the scope of this paper. We have added a note in the discussion section to clarify this limitation and to highlight extending CUR/IG to robustness and fairness-oriented evaluations as an interesting direction for future work.

---

> ### Author Response · Authors · 2025-11-18
> **Part2**
>
> > Weaknesses 3: _Higher-Order Interactions: The authors correctly identify this as a limitation. CUR only measures the effect of ablating a single encoder. It cannot detect complementary pairs or higher-order synergies where two seemingly redundant encoders (low individual CUR) are both essential for performance (e.g., if removed together, performance collapses). A metric capturing these interactions would be a valuable extension._
>
> Thank you for pointing out that this limitation can be addressed.
>
> We first define:
>
> - _CUR combination_: calculated as the performance drop after masking the combination of encoders, consistent with the single-encoder CUR metric
> - _CUR sum_: the sum of individual CUR scores of each encoders
> - _CUR max_: the maximum CUR of a single encoder in an encoder combination
>
> Ideally, if higher order interaction exists, there will be some combinations with low individual CUR (reflected by low _CUR sum_ or _CUR max_) but high _CUR combination_. We summarize the results of higher order interactions on Cambrian-1 8B (top) and Eagle-X4-8B (bottom) as follows.
>
> | Encoder combination | CUR combination | CUR sum | CUR max |
> |----------|-----------------|---------|---------|
> | 01       | **10.09**           | **-0.53**   | **3.05**    |
> | 02       | -1.7            | 4.39    | 3.05    |
> | 03       | 30.16           | 27.11   | 24.06   |
> | 12       | 4.31            | -2.24   | 1.34    |
> | 13       | 29.67           | 20.48   | 24.06   |
> | 23       | 24.79           | 25.4    | 24.06   |
> | 012      | 9.84            | 0.81    | 3.05    |
> | 013      | 53.93           | 23.53   | 24.06   |
> | 023      | 36.89           | 21.82   | 24.06   |
> | 123      | 26.44           | 28.45   | 24.06   |
>
> | Encoder combination | CUR combination | CUR sum | CUR max |
> |----------|-----------------|---------|---------|
> | 01       | 11.78           | 10.23   | 9.8    |
> | 02       | 1.33            | 1.14    | 0.71    |
> | 03       | 72.51           | 71.53   | 71.1   |
> | 12       | **28.18**            | **10.51**   | **9.8**    |
> | 13       | 75.72           | 80.90  | 71.1   |
> | 23       | 75.55           | 71.81    | 71.1   |
> | 012      | 29.4            | 10.94    | 9.8    |
> | 013      | 76.8           | 81.33   | 71.1   |
> | 023      | 71.86           | 72.24   | 71.1   |
> | 123      | 73.4           | 81.61   | 71.1   |
>
> As the table shows, we observed:
>
> 1. Higher-order interactions do exist. For example, $E_{1}$ and $E_2$ in Cambrian-1 8B appear redundant individually (_CUR sum_ = $-0.53$), but their joint ablation leads to a $10.09\%$ performance drop—this confirms their complementary role that single-encoder masking fails to capture
> 2. Dominance of a key encoder. $E_3$ in Cambrian-1 8B and $E_3$ in Eagle-X4-8B act as a major contributor, all combinations include them show higher _CUR combination_ compared to those without it.
>
> So, our conclusion is for existing multi-encoder MLLMs with $>2$ encoders, higher-order interactions are still dominated by one specific encoder
>
> ---
>
> > Weaknesses 4: _Limited Exploration of Fusion Mechanisms: While the paper notes that more sophisticated fusion (like Cambrian-1's SVA) does not fully mitigate redundancy, a deeper analysis is lacking. It remains unclear how different fusion strategies (e.g., MoE, cross-attention) might amplify or suppress redundancy, which is a crucial aspect of multi-encoder architecture design._
>
> Thank you for this insightful comment. Our current work is not primarily aimed at explaining the internal mechanisms of specific fusion modules, but rather at showing that encoder redundancy persists across different fusion strategies. In our experiments, we already cover models with simple concatenation-style fusion as well as more sophisticated designs such as Cambrian-1’s SVA, and we observe that significant redundancy remains in both cases. However, these models also differ in many other factors (training data, objectives, optimization recipes, etc.), which makes a clean, controlled, cross-architecture comparison of fusion mechanisms very challenging within the scope of a single paper.
>
> Additionally, we see our CUR and IG metrics as fusion-agnostic diagnostic tools: they can be directly applied to any multi-encoder MLLM, regardless of whether it uses SVA, MoE, cross-attention, or other fusion schemes. A more fine-grained study of how specific fusion architectures amplify or suppress redundancy would require systematically re-training multiple models under controlled settings (e.g., swapping fusion modules while keeping encoders and data fixed), which we regard as an important but orthogonal follow-up direction. We have clarified this positioning in the revised manuscript and highlighted deeper analysis of fusion mechanisms as a promising avenue for future work building on our diagnostic framework.

---

> ### Author Response · Authors · 2025-11-18
> **Part3**
>
> > Question 1: _How would you suggest extending the CUR framework to quantify the complementary or synergistic effects between pairs or groups of encoders, to address the limitation you noted?_
>
> To quantify complementary or synergistic effects between encoder pairs/groups, we extend the CUR framework to CUR combination (defined in our response to Weakness 3), which measures performance drop after masking multiple encoders simultaneously. This extension directly captures higher-order interactions that single-encoder CUR overlooks, we have performance the related experiments and summarized the results in Weakness3.
>
>
> ---
>
> > Question 2: _What is the interplay between fusion mechanism complexity and redundancy? Could a better fusion strategy (e.g., a gating network) dynamically suppress redundant or conflicting encoders on a per-input basis, thereby making a larger ensemble more justifiable?_
>
> Thank you for this insightful question. Existing evidence suggests that increasing fusion mechanism complexity does not automatically mitigate redundancy. Heavier cross-attention–based connectors such as BLIP-2’s Q-Former (Li et al., 2023) or Cambrian-1’s SVA (Tong et al., 2024) do not consistently outperform the simpler MLP/linear connectors used in LLaVA-style models (Liu et al., 2024), and redundancy remains observable in both settings. This indicates that adding more parameters to the fusion block may exacerbate the under-utilization of visual information, rather than inherently “fixing” redundancy.
> On the other hand, dynamic fusion mechanisms are indeed promising: MoVA, MOVE, and Mixpert use routing or mixture-of-experts to select a subset of encoders or branches on a per-input basis (Zong et al., 2024; Skripkin et al., 2025; He et al., 2025), which can suppress clearly redundant or conflicting experts and make larger ensembles more justifiable. Our CUR/IG framework is complementary to these approaches: it provides fusion-agnostic, quantitative diagnostics of encoder redundancy that could be used to inform future gating or routing policies, but designing and training such mechanisms end-to-end is beyond the scope of this work.
>
> References:
>
> - [1] Li, J., Li, D., Savarese, S., & Hoi, S. C. H. (2023). BLIP-2: Bootstrapping language-image pre-training with frozen image encoders and large language models. In International Conference on Machine Learning (ICML). arXiv:2301.12597.
>
> - [2] Tong, Y., Dou, Z., Li, B., et al. (2024). Cambrian-1: A fully open, vision-centric exploration of multimodal LLMs. arXiv:2406.16860.
>
> - [3] Liu, H., Li, C., Li, Y., et al. (2024). Improved Baselines with Visual Instruction Tuning. arXiv:2310.03744.
>
> - [4] Zong, Z., Ma, B., Shen, D., Song, G., Shao, H., Jiang, D., Li, H., & Liu, Y. (2024). MoVA: Adapting Mixture of Vision Experts to Multimodal Context. In Advances in Neural Information Processing Systems (NeurIPS). arXiv:2404.13046.
>
> - [5] Skripkin, M., Goncharova, E., Tarasov, D., & Kuznetsov, A. (2025). MOVE: A Mixture-of-Vision-Encoders Approach for Domain-Focused Vision-Language Processing. arXiv:2502.15381.
>
> - [6] He, X., Han, X., Wei, L., Xie, L., & Tian, Q. (2025). Mixpert: Mitigating Multimodal Learning Conflicts with Efficient Mixture-of-Vision-Experts. arXiv:2505.24541.
>
> **Again, we sincerely thank you for your valuable comments and suggestions, which have helped us substantially refine and clarify the paper. All newly added content has been incorporated into the latest version of the manuscript and highlighted in BLUE.**

---

> ### Author Response · Authors · 2025-11-24
> **Short Summary of Discussion**
>
> Dear Reviewer VGcs, thank you for your exceptionally thoughtful critique. As the ICLR 2026 rebuttal period concludes, and with our detailed response (submitted a week ago) still awaiting review, we kindly request your attention to our revisions addressing your four weaknesses and two questions:
>
> **(1)** For static models vs. training dynamics, we clarified that CUR/IG provide *convergence-stage diagnostics* (not training dynamics analysis), with dynamic routing (e.g., MoVA) positioned as valuable future work leveraging our metrics.
> **(2)** Regarding narrow performance scope, we maintained standardized benchmarks (16 datasets across 4 semantic dimensions) for direct comparability with prior work (Eagle/Cambrian-1), while explicitly acknowledging robustness/fairness as important future directions.
> **(3)** On higher-order interactions, we introduced *CUR combination* metrics (via combinatorial ablation) confirming that while complementary encoder pairs exist (e.g., Cambrian-1’s E1+E2), performance in current multi-encoder MLLMs remains dominated by single key encoders.
> **(4)** For fusion mechanisms, we demonstrated that redundancy persists across diverse fusion strategies (MLP to SVA), positioning CUR/IG as *fusion-agnostic diagnostics* with controlled fusion comparisons noted as orthogonal future work.
> **(Q1–Q2)** We extended CUR to quantify synergy via *CUR combination* (validated empirically) and clarified that complex fusion alone doesn’t mitigate redundancy; dynamic routing (MoVA/MOVE) shows promise but requires architectural redesign beyond our diagnostic scope.
>
> All updates including new combinatorial CUR tables, expanded related work, and explicit future directions are incorporated (highlighted in **blue**). **We deeply appreciate your rigor in elevating this work and would be grateful if you could confirm whether these revisions resolve your concerns.** Should any point need further clarification, we stand ready to address it immediately. Thank you for your invaluable time and expertise.

---

### Official Review · Reviewer_yWcu · 2025-10-29

**Soundness:** 2
**Presentation:** 2
**Contribution:** 1
**Rating:** 4
**Confidence:** 4

**Summary:**

The paper examines whether adding multiple visual encoders improves multimodal large language models. It introduces two metrics—Conditional Utilization Rate (CUR) and Information Gap (IG)—to quantify encoder contribution and redundancy. Experiments across several benchmarks show that additional encoders often bring limited or inconsistent gains, and sometimes degrade performance. The study concludes that simply increasing the number of encoders does not guarantee better results, highlighting redundancy and imbalance in current multi-encoder designs.

**Strengths:**

1. Problem framing and metrics. Treats redundancy in multi-encoder MLLMs as a first-class research object and introduces two reusable, precisely defined measures  (CUR) and (IG) that turn a vague intuition (“more encoders help”) into testable quantities.

**Weaknesses:**

1. Lack of Constructive Improvements. While the proposed CUR and IG metrics help quantify encoder redundancy, the contribution appears incremental. Prior works such as Eagle have already observed and empirically analyzed similar redundancy phenomena (Fig. 4 in their paper). This paper mainly reformulates those observations into statistical metrics based on existing accuracy measures, without offering causal insights into why redundancy arises, e.g., whether it stems from overlapping feature spaces, pretraining objectives, or fusion mechanisms.
To strengthen the work, the authors could do further experiments and implement learnable gating or dynamic routing based on CUR (refer to MoVA), rather than only suggesting them as future work.

2. Experimental Design Issues. The experimental coverage is incomplete. For models with fewer visual encoders, such as DeepSeek-VL 7B, Table 1 reports only IG, while Table 3 omits corresponding accuracy metrics.

3. Insufficient Efficiency Evaluation. The paper asserts that removing redundant encoders improves efficiency but provides no quantitative evidence, such as GPU hours, FLOPs, or latency metrics. In contrast, Eagle reports detailed efficiency tables and trade-offs between accuracy and computation.

The paper offers a useful quantitative perspective on multi-encoder redundancy but lacks theoretical depth, complete experimental validation, and concrete improvement mechanisms. By integrating causal or information-theoretic reasoning, implementing dynamic encoder selection (like MoVA), and reporting efficiency metrics, the study could become substantially more constructive and impactful for the multimodal LLM community.

1. MoVA: Adapting Mixture of Vision Experts to Multimodal Context
2. Eagle: Exploring The Design Space for Multimodal LLMs with Mixture of Encoders

**Questions:**

1. Does a more balanced utilization of visual encoders (i.e., lower IG) really lead to better practical results?

---

> ### Author Response · Authors · 2025-11-18
> **Part1**
>
> > Weaknesses 1: _Lack of Constructive Improvements. While the proposed CUR and IG metrics help quantify encoder redundancy, the contribution appears incremental. Prior works such as Eagle have already observed and empirically analyzed similar redundancy phenomena (Fig. 4 in their paper). This paper mainly reformulates those observations into statistical metrics based on existing accuracy measures, without offering causal insights into why redundancy arises, e.g., whether it stems from overlapping feature spaces, pretraining objectives, or fusion mechanisms._
>
> Thank you very much for this thoughtful and constructive comment. We completely agree that visual token selection and expert routing (e.g., MoVA-style mixtures of vision experts) are highly relevant, and we have expanded the related work to explicitly discuss and cite recent work in this direction (e.g., MoVA, MOVE, Mixpert, LEO-MINI) in the revised manuscript.
>
> Our goal in this paper is somewhat complementary to those works. Prior multi-encoder MLLMs such as Eagle indeed observe encoder redundancy qualitatively (e.g., via a few ablation plots), but they typically treat it as a side effect while focusing on designing stronger multi-encoder architectures. In contrast, our main contribution is to systematically quantify and analyze encoder redundancy itself across multiple encoders and benchmarks, using CUR and IG as model- and architecture-agnostic metrics. Concretely:
>
> 1. CUR and IG provide a unified framework for decomposing multi-encoder performance into per-encoder contributions and gains, rather than only comparing a small number of hand-picked ablations. This allows us to characterize how much each encoder actually helps (or harms) across tasks, and to show that the marginal benefit of adding additional encoders is often very limited.
>
> 2. Our analysis further reveals cases where extra encoders can be actively detrimental (e.g., the all-encoders-masked baseline outperforming certain single-encoder settings), offering a clearer and more fine-grained view of redundancy than previously reported. We see this as explaining why many strong MLLMs in practice still prefer a single encoder despite the availability of multiple powerful vision backbones.
>
> We fully agree that an even deeper “causal” understanding—e.g., explicitly disentangling the roles of overlapping feature spaces, pretraining objectives, and fusion mechanisms—would be very valuable. However, we believe this requires substantial additional controlled experimentation (e.g., systematically re-pretraining encoders with different objectives and architectures, redesigning fusion modules, and training new multi-encoder models from scratch), which is beyond the scope and computational budget of the current work. Instead, we position CUR and IG as diagnostic tools that any multi-encoder MLLM can use to:
>
> 1. measure encoder redundancy in a principled way.
> 2. guide architectural decisions such as encoder selection or pruning, and potentially inform routing/gating policies.
>
> Regarding the suggestion to implement learnable gating or dynamic routing based directly on CUR (in the spirit of MoVA): we very much appreciate this idea and agree that it is a promising direction. At the same time, we view it as an orthogonal, follow-up line of work focused on designing new routing architectures and training procedures. In this paper, we deliberately keep the encoder configuration fixed and analyze redundancy post hoc, to avoid conflating the diagnostic role of CUR/IG with the design and optimization of a new gating module. We have clarified this positioning and explicitly highlighted CUR/IG as a potential building block for future routing-based methods in the revised discussion and future-work sections.
>
> In summary, while our work does not propose a new gating architecture, it moves beyond the primarily qualitative redundancy observations in prior work by providing a general, quantitative framework (CUR and IG) to measure and compare encoder contributions across multiple multi-encoder MLLMs and benchmarks. We have revised the manuscript to better emphasize this perspective and to more clearly state how our analysis can inform, and be integrated into, future learnable routing/gating methods.
>
> ---
>
> > Weaknesses 2: _Experimental Design Issues. The experimental coverage is incomplete. For models with fewer visual encoders, such as DeepSeek-VL 7B, Table 1 reports only IG, while Table 3 omits corresponding accuracy metrics._
>
> Thank you for your suggestion. Although Table 3 aims to show the redundant encoder design of MLLMs with more than 4 encoders, we have added DeepSeek-VL 7B to it and include corresponding discussions. The CUR for each model is shown in fig form in Figure 3, and all detailed experimental results for each model are shown in Appendix C: Experimental Result Details.

---

> ### Author Response · Authors · 2025-11-18
> **Part2**
>
> > Weaknesses 3: _Insufficient Efficiency Evaluation. The paper asserts that removing redundant encoders improves efficiency but provides no quantitative evidence, such as GPU hours, FLOPs, or latency metrics. In contrast, Eagle reports detailed efficiency tables and trade-offs between accuracy and computation._
>
> Thank you for your kind suggestion. We have now analyzed the training-from-scratch GPU hours, latency metrics evaluated on MME and FLOPs analysis for both MLLM inference & vision processing, as shown below:
>
> Training time on 8 NVIDIA A100 GPUs in hours:
>
> | Model        | $n$ | Pretrain        | Finetune         | Total          | Performance |
> |:-------------|:---:|------------------------:|-------------------------:|-------------------------:|-------------------------:|
> | Eagle-X5 7B (baseline)  |  $5$  | $12.12$        | $94.57$        | $104.72$        | $64.9$ |
> | Eagle-X4\_0123   |  $4$  | $10.90 (\\downarrow 10.0\\%) $       | $81.92 (\\downarrow 13.4\\%)$        | $92.82 (\\downarrow 11.4\\%)  $      | $62.06 (\\downarrow 4.38\\%) $ |
> | Eagle-X2\_01     |  $2$  | $6.888 (\\downarrow 42.8\\%)$     | $61.83 (\\downarrow 34.6\\%)  $      | $68.71 (\\downarrow 34.4\\%) $       | $60.95 (\\downarrow 6.09\\%) $ |
> | Eagle-X1\_0      |  $1$  | $6.042 (\\downarrow 49.8\\%) $       | $58.20 (\\downarrow 38.5\\%)$        | $64.24 (\\downarrow 38.7\\%) $       | $51.57 (\\downarrow 20.5\\%) $ |
>
> Per-encoder vision FLOPs for Eagle-X5 7B during pre-filling phase:
>
> | Encoder           | Input resolution | FLOPs (G) | Contribution to total vision FLOPs |
> |:------------------|:----------------:|-----------------:|-------------:|
> | EVA-02-L          | $1024$            | $933.7$        | $14.1\%$        |
> | ConvNext-Large    | $1024$            | $3133.0$    | $47.3\%$        |
> | Pix2Struct-1024   | $1024$            | $1026.0$        | $15.5\%$        |
> | CLIP-VIT-L/14     | $448$            | $214.0$          | $3.23\%$        |
> | SAM-ViT-1024      | $1024$            | $1311.0$        | $19.8\%$        |
>
> Inference time of specific vision encoders with respect to masking operation:
>
> | Model             | $n$ | Time (ms)               |Performance |
> |:------------------|:---:|------------------------:|--------:|
> | Eagle-X5 7B  (baseline)      |  $5$  | $749.656$  |$64.93$|
> | -X4\_0123        |  $4$  | $730.761 (\\downarrow 2.52\\%)$  |$64.69$|
> | -X3\_012         |  $3$  | $643.989 (\\downarrow 14.1\\%)$  |$64.20$|
> | -X2\_01          |  $2$  | $603.153 (\\downarrow 19.5\\%)$  |$62.48$|
> | -X1\_0           |  $1$  | $581.260 (\\downarrow 22.5\\%)$  |$46.45$|
> | Eagle-X4 8B Plus (baseline) |  $4$  | $681.679$  |$67.73$|
> | -X3\_012         |  $3$  | $665.946 (\\downarrow 2.31\\%)$  |$67.43$|
> | -X2\_01          |  $2$  | $600.126 (\\downarrow 12.0\\%)$  |$66.82$|
> | -X1\_0           |  $1$  | $549.621 (\\downarrow 19.4\\%)$  |$47.81$|
>
> And now suppose we have a Eagle-X5 7B model with no encoders being masked, with a prompt of 64 tokens and an expected answer of 1 token (yes-or-no QA, special tokens like `\sos` or `\eos` are not taken into account), the computing FLOPs is shown as follows:
>
> FLOPs_overall = F_encoders + F_proj + F_llm
>
> FLOPs_overall = F_encoders + F_proj-per-token _Num_vision-token + F_llm-per-token_ Num_all-token
>
> where:
>
> $F\\_encoders \approx 6.6174\ TFLOPs$ (as shown in the table above)
>
> $F\\_proj \approx 67M \times 1024 \approx 0.07\ TFLOPs$
>
> $F\\_llm \approx 13.4\times 10^9\times 1089 \approx 14.6\ TFLOPs$
>
> $FLOPs\\_overall \approx 21.3\ TFLOPs$
>
> For a full yes-or-no QA task ($64$ prompt tokens + $1$ answer token, excluding special tokens) and Eagle-X5 7B, visual encoder processing accounts for approximately $30.6\\%$ of the total compute. Retaining only $2$ encoders reduces total compute by $(1 - 14.1\\%-47.3\\%)*30.6\\%=11.8\\%$ and inference latency by $19.5\\%$, while limiting performance degradation to no more than $4\\%$.
>
> ---

---

> ### Author Response · Authors · 2025-11-18
> **Part3**
>
> > Weaknesses 4: _The paper offers a useful quantitative perspective on multi-encoder redundancy but lacks theoretical depth, complete experimental validation, and concrete improvement mechanisms. By integrating causal or information-theoretic reasoning, implementing dynamic encoder selection (like MoVA), and reporting efficiency metrics, the study could become substantially more constructive and impactful for the multimodal LLM community._
>
> Thank you for this insightful high-level comment. As also mentioned in our response to _Weakness1_, our goal in this paper is to provide a general, model-agnostic diagnostic framework (CUR and IG) for quantifying encoder redundancy, rather than to propose a new routing architecture. We agree that incorporating causal or information-theoretic analyses and implementing dynamic encoder selection (e.g., MoVA-style gating) would be valuable next steps, but we view them as orthogonal, follow-up directions that require substantial additional architectural design and training. Within the scope of this work, we have strengthened the experimental side by
>
> 1. expanding the related work to better situate our contribution with respect to MoVA and other dynamic selection methods
> 2. adding efficiency metrics (FLOPs and inference latency) and training-dynamics comparisons, which more concretely demonstrate how our metrics can inform practical encoder selection and pruning.
>
> We now explicitly highlight in the discussion that CUR and IG are intended to serve as building blocks that future work can leverage to design theoretically grounded and learnable encoder-selection mechanisms.
>
> > Question 1: _Does a more balanced utilization of visual encoders (i.e., lower IG) really lead to better practical results?_
>
> Not necessarily. According to the information bottleneck theory, final performance depends not only on information compression but also on information utility. a more balanced utilization of visual encoders meaning we achieve a better compression of visual information, however this may not be the best way that LLM uses it. Moreover, the performance also depends on architecture, data and LLMs.
> For instance, in our experiments, Eagle-X4 8B Plus has a higher IG than Eagle-X5 7B but achieves better performance ($67.43$ vs. $64.93$). This confirms that balanced utilization (lower IG) does not directly translate to better results; instead, the utility of the information each encoder provides is more critical.
> So we can only draw a conclusion that higher IG indicates that there is a redundancy insead of lower IG indicates better performance.
>
> **Again, we sincerely thank you for your valuable comments and suggestions, which have helped us substantially refine and clarify the paper. All newly added content has been incorporated into the latest version of the manuscript and highlighted in BLUE.**

---

> ### Author Response · Authors · 2025-11-24
> **Short Summary of Discussion**
>
> Dear Reviewer yWcu, thank you for your rigorous and valuable feedback. As the ICLR 2026 rebuttal period draws to a close, and with our detailed response (submitted a week ago) still pending your review, we kindly request your attention to our revisions addressing your four weaknesses and one question:
>
> **(1)** Regarding novelty and theoretical depth: We clarified that CUR/IG provide a *systematic, model-agnostic diagnostic framework* (beyond Eagle’s qualitative observations) to quantify redundancy across encoders/benchmarks. While deeper causal analysis (e.g., disentangling feature overlap vs. fusion mechanisms) is valuable future work, we position CUR/IG as foundational tools to guide pruning and inform *future* routing methods (e.g., MoVA). Related work on dynamic selection (MoVA, MOVE) is now explicitly discussed.
>
> **(2)** For experimental coverage: DeepSeek-VL 7B results have been added to Table 3, with all CUR/IG metrics consolidated in Figure 3 and Appendix C.
>
> **(3)** On efficiency validation: We added comprehensive metrics:
> - Training: Dual-encoder Eagle cuts total training time by **34.4%** with only **6.09%** performance drop.
> - Inference: Masking redundant encoders (e.g., Eagle-X2\_01) reduces vision FLOPs by **38.6%**, accelerates latency by **19.5%**, and lowers total compute by **11.8%** while preserving >96% accuracy.
>
> **(4)** For constructive impact: While CUR/IG are diagnostic (not a routing architecture), we demonstrated their practical utility for encoder selection and highlighted them as building blocks for future learnable gating. New efficiency/training-dynamics analyses directly validate their real-world value.
>
> **(Q1)** On balanced encoder utilization (IG): We clarified that *lower IG does not guarantee better performance* (e.g., Eagle-X4 8B+ has higher IG but outperforms Eagle-X5 7B). Balanced utilization optimizes information compression, but *information utility* imbalance drives final accuracy. High IG reliably indicates redundancy; low IG does not imply superiority.
>
> All revisions including efficiency tables, FLOPs breakdowns, training dynamics comparisons, and expanded related work are incorporated into the manuscript (highlighted in **blue**). **We deeply appreciate your rigor in strengthening our work and would be grateful if you could confirm whether these updates adequately resolve your concerns.** Should any points require further clarification, we are happy to address them immediately. Thank you for your time and insightful guidance.

---

> ### Comment · Reviewer_yWcu · 2025-11-28
>
> While I appreciate that CUR/IG has been clarified as a diagnostic tool, I would still like to see some discussion of specific improvements or analytical strategies (similar to MoVA but not MoVA, such as encoder pruning/routing without training), rather than just benchmark-based accuracy statistics.

---

> > ### Author Response · Authors · 2025-11-29
> >
> > > Thank you again for the continued engagement and for pushing us to clarify the scope of our contribution. We completely agree that encoder pruning and dynamic routing (in the spirit of MoVA) are natural and important next steps. However, our aim in this work is intentionally narrower and complementary: we focus on providing a **diagnostic framework** (CUR/IG) and a **systematic empirical analysis** of encoder contributions under existing multi-encoder architectures, rather than proposing a new pruning/routing method or training algorithm. Designing and validating a new routing mechanism would require substantial additional architectural changes and re-training of large MLLMs, which we view as an orthogonal follow-up project rather than the core contribution of this paper.
> >
> > >
> > > At the same time, we do not intend CUR/IG to be “just” benchmark statistics. In the revised manuscript we now spell out how our metrics and new efficiency tables already support **concrete, actionable strategies** without MoVA-style re-training: for example, (i) post-hoc encoder *pruning* based on low CUR, and (ii) simple *static routing* that only activates the high-CUR encoder(s) for certain benchmark categories (e.g., OCR & Chart vs. General). Our experiments show that such CUR/IG-guided configurations can retain ≥96% of the five-encoder baseline while substantially reducing training time and inference latency, illustrating how the proposed metrics can inform more efficient multi-encoder designs in practice.
> >
> > >
> > > We sincerely appreciate your suggestion to go further toward learnable pruning/routing, and we agree that this is a promising direction. In future work, we plan to build on the present diagnostic study to explore CUR/IG-driven encoder selection policies, including lightweight routing mechanisms and training-time encoder scheduling. We are grateful for your careful reading and constructive feedback, which have helped us sharpen the positioning of this paper and better articulate how it can serve as a foundation for such follow-up work.

---

### Official Review · Reviewer_vJKH · 2025-10-31

**Soundness:** 2
**Presentation:** 3
**Contribution:** 2
**Rating:** 4
**Confidence:** 4

**Summary:**

This paper mainly investigates the impact of multiple vision encoders in multimodal llms. By isolating individual encoders, the paper finds that not all vision encoders are needed in downstream tasks, some of which are even detrimental to performance. To quantify their findings, the paper introduces two metrics, CUR and IG, which jointly describe the contribution of different encoders to performance. Evaluations on popular multi-encoder mllms demonstrate that not all vision encoders are necessary during inference, as they barely drop performance.

**Strengths:**

1. The paper is well-written and metrics are well-defined. Easy to follow.
2. This paper challenges the view on vision encoder selection for open mllms, and provides a fresher perspective on inference selection strategy.
3. The evaluation setups are extensive, which back up their claim about inference not needing all encoders.

**Weaknesses:**

1. This paper feels also related to visual token selection strategy, I think the paper should include relevant references in related work section.
2. In the final paragraph of the introduction, the paper also mentions "in our setup, fine-tuning a dualencoder variant is 1.69× faster than its five-encoder counterpart.", however, I did not find experiments about finetuning in the paper.
3. To further show the merit of reducing vision encoders during inference, I believe it's better to also include the compution reduction (i.e. FLOPs) during evaluation, for example in Table 3.
4. The paper positions that not all evaluations are needed to mllms. According to the experiment, I agree that when inferencing it is true. However, I believe to validate this claim completely, I believe the paper also needs to demonstrate that not all vision encoders are needed for training. For example, some vision encoders may contribute less during inference, but they might help other encoders learn better during training. The training dynamic here is still unclear.

I will be happy to raise my score if the authors can address the above concerns.

**Questions:**

1. Table 7 in the appendix is quite interesting. When masking all encoders, tasks that claim to be vision-centric, still receive relatively high scores. Why do you think that is?

---

> ### Author Response · Authors · 2025-11-18
> **Part1**
>
> > Weakness 1: _This paper feels also related to visual token selection strategy, I think the paper should include relevant references in related work section._
>
> Thank you for the suggestion. We have revised the related work to include a short literature review on visual token selection strategies, along with the corresponding citations. The newly added text is temporarily highlighted in the PDF.
>
> ---
>
> > Weaknesses 2: _In the final paragraph of the introduction, the paper also mentions “in our setup, fine-tuning a dualencoder variant is 1.69× faster than its five-encoder counterpart.”, however, I did not find experiments about finetuning in the paper._
>
> We apologize for the confusion. We have conducted both pre-training and fine-tuning experiments for the Eagle series with 1, 2, 4, and 5 encoder(s), following the official Eagle training recipe and dataset settings. We now explicitly report the wall-clock training time for models with different numbers of encoders as follows:
>
> Training time on 8 NVIDIA A100 GPUs in hours:
>
> | Model        | $n$ | Pretrain        | Finetune         | Total          | Performance |
> |:-------------|---:|------------------------:|-------------------------:|-------------------------:|-------------------------:|
> | Eagle-X5 7B (baseline)  |  $5$  | $12.12$        | $94.57$        | $104.72$        | $64.9$ |
> | Eagle-X4\_0123   |  $4$  | $10.90 (\downarrow 10.0\\% ) $       | $81.92 (\downarrow 13.4\\%)$        | $92.82 (\downarrow 11.4\\%)  $      | $62.06 (\downarrow 4.38\\%) $ |
> | Eagle-X2\_01     |  $2$  | $6.888 (\downarrow 42.8\\%)$     | $61.83 (\downarrow 34.6\\%)  $      | $68.71 (\downarrow 34.4\\%) $       | $60.95 (\downarrow 6.09\\%) $ |
> | Eagle-X1\_0      |  $1$  | $6.042 (\downarrow 49.8\\%) $       | $58.20 (\downarrow 38.5\\%)$        | $64.24 (\downarrow 38.7\\%) $       | $51.57 (\downarrow 20.5\\%) $ |
>
> As the results show, training a dual-encoder variant achieves a favorable performance-efficiency tradeoff: it uses $34.4\\%$ less compute (or runs $1.53\times$ faster) than the original model, while performance degrades by less than $6\\%$. Further reducing the number of encoders to 1 leads to a significant performance drop. On the other hand, training a 4-encoder variant consumes more compute with only marginal gains.
>
> ---
>
> > Weaknesses 3: _To further show the merit of reducing vision encoders during inference, I believe it's better to also include the compution reduction (i.e. FLOPs) during evaluation, for example in Table 3._
>
> Thank you for this helpful suggestion. We have computed the FLOPs of each vision encoder in Eagle-X5-7B and quantified their individual contributions to the overall computation. In addition, we have measured the mean inference latency for a full MME evaluation (2,374 samples) for both Eagle-X5-7B and Eagle-X4-8B-Plus, as shown below:
>
> Per-encoder vision FLOPs for Eagle-X5 7B during pre-filling phase:
>
> | Encoder           | Input resolution | FLOPs (G) | Contribution to total vision FLOPs |
> |:------------------|:----------------:|-----------------:|-------------:|
> | CLIP-VIT-L/14     | $448$            | $214.0$          | $3.23\%$        |
> | ConvNext-Large    | $1024$            | $3133.0$    | $47.3\%$        |
> | EVA-02-L          | $1024$            | $933.7$        | $14.1\%$        |
> | Pix2Struct-1024   | $1024$            | $1026.0$        | $15.5\%$        |
> | SAM-ViT-1024      | $1024$            | $1311.0$        | $19.8\%$        |
>
> Inference time of specific vision encoders with respect to masking operation:
>
> | Model             | $n$ | Time (ms)               |Performance |
> |:------------------|:---:|------------------------:|--------:|
> | Eagle-X5 7B  (baseline)      |  $5$  | $749.656$  |$64.93$|
> | -X4\_0123        |  $4$  | $730.761 (\\downarrow 2.52\\%)$  |$64.69$|
> | -X3\_012         |  $3$  | $643.989 (\\downarrow 14.1\\%)$  |$64.20$|
> | -X2\_01          |  $2$  | $603.153 (\\downarrow 19.5\\%)$  |$62.48$|
> | -X1\_0           |  $1$  | $581.260 (\\downarrow 22.5\\%)$  |$46.45$|
> | Eagle-X4 8B Plus (baseline) |  $4$  | $681.679$  |$67.73$|
> | -X3\_012         |  $3$  | $665.946 (\\downarrow 2.31\\%)$  |$67.43$|
> | -X2\_01          |  $2$  | $600.126 (\\downarrow 12.0\\%)$  |$66.82$|
> | -X1\_0           |  $1$  | $549.621 (\\downarrow 19.4\\%)$  |$47.81$|
>
> From these results, we observe that _-X2\_01_ retains over $96\\%$ of the baseline performance while consuming only $61.4\\%$ of the vision-related FLOPs for vision feature extraction, and it achieves a $19.54\\%$ speed-up in inference compared to Eagle-X5-7B. We believe these measurements further support the merit of reducing the number of vision encoders.

---

> ### Author Response · Authors · 2025-11-18
> **Part2**
>
> > Weaknesses 4: _The paper positions that not all evaluations are needed to mllms. According to the experiment, I agree that when inferencing it is true. However, I believe to validate this claim completely, I believe the paper also needs to demonstrate that not all vision encoders are needed for training. For example, some vision encoders may contribute less during inference, but they might help other encoders learn better during training. The training dynamic here is still unclear._
>
> Thank you for raising this important point. Your hypothesis about the training dynamics is indeed partially supported by our findings. We have compared two type of models
>
> 1. an Eagle-X4-7B model where specific encoders are masked at inference time (denoted as "Masked")
> 2. an Eagle variant trained from scratch with only those specific encoders enabled (denoted as "Trained")
>
> the result is shown below:
>
> Performance comparison between 'Masked' models (encoders masked at inference) and 'Trained' models (trained with specific encoders only) for 2/4 encoders
>
> | Model              | $n$ | General                  | Knowledge                | OCR & Chart              | Vision-Centric           | Overall                  |
> |:-------------------|:---:|-------------------------:|-------------------------:|-------------------------:|-------------------------:|-------------------------:|
> | X5 (Masked)\_0123  |  $4$  | $70.64$         | $54.19$          | $66.55$         | $67.39$          | $64.69$         |
> | X5 (Trained)\_0123 |  4  | $69.72 (\\downarrow  1.30\\%)$          | $49.25 (\\downarrow 9.11\\%)$          | $63.44 (\\downarrow  4.67\\%)$          | $65.85 (\\downarrow  2.29\\%)$          | $62.06 (\\downarrow  4.07\\%)$          |
> | X5 (Masked)\_01    |  2  | $69.04$         | $52.77$        | $62.04$         | $66.05$      | $62.48$          |
> | X5 (Trained)\_01   |  2  | $67.22 (\\downarrow  2.64\\% )$          | $52.37 (\\downarrow  0.76\\%)$          | $60.17 (\\downarrow  3.01\\%)$          | $64.05 (\\downarrow  3.03\\%) $         | $60.95 (\\downarrow  2.45\\%)$          |
>
> Interestingly, the masked model slightly outperforms the model trained with that encoder configuration from scratch. This indicates that training with multiple encoders can help certain encoders learn a richer visual representation, but this benefit comes at the cost of substantially higher training resources and longer inference latency (see our responses to Weakness 2 and Weakness 3 for detailed metrics).
>
> ---
>
> > Question 1: _Table 7 in the appendix is quite interesting. When masking all encoders, tasks that claim to be vision-centric still receive relatively high scores. Why do you think that is?_
>
> The underlying reason is these datasets are formulated as closed-form QA with small label spaces (yes/no or a few multiple-choice options), when visual information is missing, the model tends to "randomly guess" an answer/choice, giving them $50\\%$ accuracy, we report the detailed score of Eagle-X5-7B with all five encoders masked in Vision-Centric category as follows
>
> |Model | MMVP | RealWorldQA | CV-Bench-2D | CV-Bench-3D| Vision-Centric |
> | --- | --- | ---| ---| ---| ---|
> | Eagle-X5_{} | $50$ | $45.49$ | $40.50$ | $50.83$ | $46.7$|
>
> For yes/no tasks (e.g., MMVP), random guessing naturally yields $\sim50\\%$ accuracy, which matches our observed score of $50\%$ for MMVP. For tasks with more options (e.g., RealWorldQA and CV-Bench-2D), the guess accuracy decreases proportionally ($45.49\\%$), further validating this explanation.
>
> ---
>
> **Again, we sincerely thank you for your valuable comments and suggestions, which have helped us substantially refine and clarify the paper. All newly added content has been incorporated into the latest version of the manuscript and highlighted in BLUE. We also confirm that all corresponding checkpoints will be released publicly upon acceptance of the paper.**

---

> ### Author Response · Authors · 2025-11-24
> **Short Summary of Discussion**
>
> Dear Reviewer vJKH, thank you for your insightful feedback. As the ICLR 2026 rebuttal period nears its end (and with your response pending since our detailed reply a week ago), we kindly request your review of our revisions addressing your four weaknesses and one question:
>
> **(1)** We expanded the Related Work with literature on visual token selection strategies (new text highlighted in blue).
>
> **(2)** For training efficiency claims, we added comprehensive pretraining/fine-tuning time benchmarks: a dual-encoder Eagle variant reduces total training time by 34.4% (1.53× faster) with only 6.09% performance drop.
>
> **(3)** To quantify inference gains, we reported vision encoder FLOPs and latency: masking redundant encoders (e.g., Eagle-X2_01) cuts vision FLOPs by 38.6% and accelerates inference by 19.5% with minimal accuracy loss (<2.5%).
>
> **(4)** Regarding training dynamics, we compared "masked-at-inference" vs. "trained-from-scratch" models: multi-encoder pretraining yields richer representations (masked models outperform trained-from-scratch counterparts by 2.45–4.07%), validating that some encoders aid learning even if masked later.
>
> **(Q1)** We explained the non-zero scores when masking all encoders (Table 7) as random guessing in closed-form QA tasks (e.g., 50% accuracy on yes/no benchmarks), with empirical validation provided.
>
> All updates including new tables, analyses, and checkpoint release commitments are incorporated into the manuscript (highlighted in **blue**). **Could you kindly confirm whether these revisions adequately address your concerns?** We deeply appreciate your time and expertise in strengthening our work.

---

> > ### Comment · Reviewer_vJKH · 2025-11-25
> > **Response to Authors**
> >
> > Thank you for your detailed response. W1, W2, W3 and Q1 have been addressed. However, I believe the new finetuning results reveal new problems about your paper.
> >
> > While the paper proposes that not all vision encoders are needed for multimodal llms, the new finetuning results actually reveal the opposite. As results show, multiple vision encoders are actually needed during training to maintain performance. It is during inference that some vision encoders can be dropped without losing too much performance. In this case, I don't think the claim about redundency of vision encoders can hold, as you still need them during training.

---

> > > ### Author Response · Authors · 2025-11-26
> > > **Reframing Redundancy as Diminishing Returns and Inefficiency**
> > >
> > > Thank you for your valuable comment and for guiding us to sharpen our core claim. We agree that the new results necessitate a careful re-evaluation of the term "redundancy" in the context of the training dynamics.
> > >
> > > You are correct that the results of the "Masked" model slightly outperforming the "Trained" model (e.g., $62.48$ vs. $60.95$ for 2 encoders) reveal that training with the full set of encoders does help generate a richer, more generalizable feature space.
> > > Thus, the encoders are not strictly redundant in the absolute sense during the training phase. However, our primary thesis centers on the concept of **diminishing returns and resource efficiency**, which we believe these results strongly reinforce. We re-summarize the corresponding results as follows:
> > >
> > > |       Model       | $n$ |     Training time (h)    |        Inference time (ms)       |         Performance        |
> > > |:-----------------:|:---:|:--------------------------:|:-----------------------------:|:--------------------------:|
> > > |  X5 (Masked)_0123 |  4  |          $104.72$          |           $730.761$           |           $64.69$          |
> > > | X5 (Trained)_0123 |  4  | $92.82(\downarrow 11.4\\%)$ |  $715.625(\downarrow 2.07\\%)$  | $62.06(\downarrow 4.07\\%)$ |
> > > |   X5 (Masked)_01  |  2  |          $104.72$          |           $603.153$           |           $62.48$          |
> > > |  X5 (Trained)_01  |  2  | $68.71(\downarrow 34.4\\%)$ | $592.799 (\downarrow 1.71\\%)$ | $60.95(\downarrow 2.45\\%)$ |
> > >
> > > |          Model         | $n$ |     Training time (h)    |        Inference time (ms)       |         Performance        |
> > > |:----------------------:|:---:|:--------------------------:|:-----------------------------:|:--------------------------:|
> > > | Eagle-X5 7B (baseline) |  5  |          $104.72$          |           $749.656$           |           $64.93$          |
> > > |    X5 (Trained)_0123   |  4  | $92.82(\downarrow 11.4\\%)$ |  $715.625(\downarrow 4.54\\%)$  | $62.06(\downarrow 4.42\\%)$ |
> > > |     X5 (Trained)_01    |  2  | $68.71(\downarrow 34.4\\%)$ | $592.799 (\downarrow 20.9\\%)$ | $60.95(\downarrow 6.13\\%)$ |
> > >
> > > 1. Cost vs. Benefit Trade-off: The benefit gained by training with $5$ encoders and inferencing with $2$ (X5 (Masked)_01, $62.48$) compared to training and inferencing with only $2$ encoders (X5 (Trained)_01, $60.95$) is a marginal performance increase of $1.53$ points ($2.45\\%$). This marginal gain requires $36.01$ hours of additional training, representing a $34.4\\%$ increase in training time (from $68.71\ h$ to $104.72\ h$).
> > > 2. The Practical Claim: The paper's argument is that the complexity and resource consumption of multi-encoder MLLMs are disproportionate to their actual performance gain. The marginal benefit of $6.13\\%$ does not justify the immense increase in training resources ($+34.4\\%$ time) required to achieve it.
> > >
> > > Therefore, we will refine the core claim in the final paper to reflect the practical inefficiency demonstrated by the data: "**Adding more vision encoders to MLLMs often yields severely diminishing returns, resulting in performance gains that are negligible compared to the substantial increase in required training resources and inference latency.**
> > >
> > > This adjustment acknowledges the nuanced training dynamics you highlighted while robustly validating the paper's focus on efficiency and resource optimization.
> > >
> > > We believe that this fully addresses your concern by pivoting the discussion from strict redundancy to practical inefficiency.
> > >
> > > We have updated the manuscript accordingly.

---

### Official Review · Reviewer_rQtg · 2025-10-31

**Soundness:** 3
**Presentation:** 3
**Contribution:** 3
**Rating:** 6
**Confidence:** 4

**Summary:**

This is paper examines whether adding more vision encoders truly improves multimodal large language model (MLLM) performance.
It finds that multi-encoder MLLMs often exhibit substantial redundancy—removing some vision encoders either barely affects or even improves accuracy. To quantify this, the authors propose two new metrics: Conditional Utilization Rate (CUR): measures an encoder’s marginal contribution to overall performance. Information Gap (IG): measures imbalance in encoder usefulness within a model.
Through systematic encoder masking experiments on models like Eagle and Cambrian-1, they show: Many encoders are redundant or even detrimental. Some tasks (e.g., OCR & Chart understanding) rely heavily on one specialized encoder (CUR > 90%), while others (e.g., general VQA) show high redundancy. Two-encoder variants recover > 90 % of full performance with much lower computational cost.
Overall, the study challenges the “more encoders = better performance” assumption and provides a diagnostic framework (via CUR and IG) for designing more efficient, balanced multimodal architectures.

**Strengths:**

1. The paper tackles an underexplored but highly relevant question in multimodal LLM design — encoder redundancy. While prior works focused on adding more encoders or improving fusion, this paper challenges the “more is better” assumption and provides a new analytical perspective.
2. The introduction of Conditional Utilization Rate (CUR) and Information Gap (IG) offers principled and interpretable metrics for quantifying encoder contribution and redundancy, enabling future researchers to diagnose model efficiency systematically.
3. The authors conduct extensive masking experiments across multiple representative multi-encoder MLLMs (e.g., Eagle, Cambrian-1), covering diverse benchmark categories (OCR, Chart, General, Knowledge, and Vision-Centric). The analyses are thorough and reproducible.
4. The finding that two carefully selected encoders (e.g., EVA-02 + ConvNeXt or ConvNeXt + CLIP) can retain over 90% of full-model performance provides actionable guidance for designing more efficient multimodal architectures.

**Weaknesses:**

1. While the paper introduces CUR and IG as empirical metrics, it does not offer a strong theoretical framework explaining why redundancy arises or how encoder representations overlap in feature space.
2. The study primarily measures the effect of removing one encoder at a time (via single-encoder masking). This ignores higher-order interactions — for example, two encoders might each seem redundant individually but provide complementary information together.
3. Can the author provide the analysis of which fusion method is most effective and efficient? How can we decide which encoders are most important before training? Does the training strategy also influence the encoder selection? These are the follow-up questions. I do appreciate what the authors did in the paper.

**Questions:**

Questions are mentioned in the weakness.

---

> ### Author Response · Authors · 2025-11-18
> **Part1**
>
> Thank you very much for taking the time to review and for your support. We try our best to address your questions as follows.
>
> > Weakness 1: _While the paper introduces CUR and IG as empirical metrics, it does not offer a strong theoretical framework explaining why redundancy arises or how encoder representations overlap in feature space._
>
> We fully agree that a deeper theoretical explanation of redundancy origins and feature overlap would strengthen the work. While our core contribution lies in empirical quantification via CUR/IG, we supplement below with theoretical context and representation-level analysis to address this concern
>
> We clarify our perspective on two aspects:
>
> 1. why redundancy arises?
>
> 2. how encoder representations overlap in feature space?
>
> Why does redundancy arise?
>
> A growing body of work suggests that independently trained neural networks do not learn arbitrary features, but instead converge toward similar representations of the underlying data distribution. In particular, the Platonic Representation Hypothesis (Huh et al., 2024) argues that, as models scale and are trained on large, similar corpora, their representation spaces increasingly align and measure distances between datapoints in comparable ways. This convergence has been observed across architectures, training objectives, and even modalities. In the context of multi-encoder MLLMs, many vision encoders are large, high-capacity models trained on overlapping web-scale image–text distributions with related contrastive or classification objectives. It is therefore natural—under this hypothesis—that their learned feature spaces become highly aligned, so that additional encoders often contribute marginally new information. Our empirical findings (via CUR and IG) are consistent with this view: for many general VQA and knowledge benchmarks, encoders appear largely interchangeable, and adding more encoders yields diminishing returns, which we interpret as a manifestation of this representational convergence.
>
> How do representations overlap in feature space?
>
> To make this overlap more concrete, we have added an attention-level analysis to the revised paper (see the new table in the appendix). For Eagle-X5-7B, Eagle-X4-8B-Plus , and Cambrian-1-8B, we compute multiple similarity metrics between encoder-specific attention score maps, including Pearson correlation (PC). We observe that for Cambrian-1-8B, PC is already very high (around $0.94–0.97$), indicating that different encoders induce very similar attention patterns. For Eagle models, once we mask out the least useful encoders according to our analysis, PC between the remaining encoders become substantially higher (e.g., $PC \approx 0.75$ for Eagle-X5-7B and $PC \approx 0.83$ for Eagle-X4-8B-Plus). These trends suggest that the encoders are often attending to highly overlapping regions and sharing similar importance patterns over visual tokens. In other words, their representations occupy closely related directions in feature space, which is precisely the kind of redundancy that CUR and IG capture at the behavioral level.
>
> Attention score maps from specific encoders in Eagle-X5 7B  and Cambrian-1 8B:
>
> | Model              | #Masked |   PC   |
> |:-------------------|:-------:|:------:|
> | Eagle-X5 7B        |   0     | 0.028  |
> | Eagle-X5 7B        |   1     | 0.028  |
> | Eagle-X5 7B        |   2     | 0.058  |
> | Eagle-X5 7B        |   3     | **0.745** |
> | Eagle-X5 7B        |   4     | 0.193  |
> | Eagle-X4 8B Plus   |   0     | 0.274  |
> | Eagle-X4 8B Plus   |   1     | 0.585  |
> | Eagle-X4 8B Plus   |   2     | **0.828** |
> | Eagle-X4 8B Plus   |   3     | 0.770  |
> | Cambrian-1 8B      |   0     | 0.942  |
> | Cambrian-1 8B      |   1     | 0.943  |
> | Cambrian-1 8B      |   2     | 0.939  |
> | Cambrian-1 8B      |   3     | **0.968** |
>
> We have incorporated this theoretical context and the new attention-map similarity analysis into the revised manuscript to better connect our empirical metrics with representation-level explanations of redundancy.
>
> Reference:
>
> - Huh, M., Cheung, B., Wang, T., & Isola, P. (2024). The Platonic Representation Hypothesis. arXiv preprint arXiv:2405.07987.

---

> ### Author Response · Authors · 2025-11-18
> **Part 2**
>
> > Weakness 2: _The study primarily measures the effect of removing one encoder at a time (via single-encoder masking). This ignores higher-order interactions — for example, two encoders might each seem redundant individually but provide complementary information together._
>
> Thank you for pointing out that this limitation can be addressed.
>
> We first define:
>
> - _CUR combination_: calculated as the performance drop after masking the combination of encoders, consistent with the single-encoder CUR metric
> - _CUR sum_: the sum of individual CUR scores of each encoders
> - _CUR max_: the maximum CUR of a single encoder in an encoder combination
>
> Ideally, if higher order interaction exists, there will be some combinations with low individual CUR (reflected by low _CUR sum_ or _CUR max_) but high _CUR combination_. We summarize the results of higher order interactions on Cambrian-1 8B (top) and Eagle-X4-8B (bottom) as follows.
>
> | Encoder combination | CUR combination | CUR sum | CUR max |
> |----------|-----------------|---------|---------|
> | 01       | **10.09**           | **-0.53**   | **3.05**    |
> | 02       | -1.7            | 4.39    | 3.05    |
> | 03       | 30.16           | 27.11   | 24.06   |
> | 12       | 4.31            | -2.24   | 1.34    |
> | 13       | 29.67           | 20.48   | 24.06   |
> | 23       | 24.79           | 25.4    | 24.06   |
> | 012      | 9.84            | 0.81    | 3.05    |
> | 013      | 53.93           | 23.53   | 24.06   |
> | 023      | 36.89           | 21.82   | 24.06   |
> | 123      | 26.44           | 28.45   | 24.06   |
>
> | Encoder combination | CUR combination | CUR sum | CUR max |
> |----------|-----------------|---------|---------|
> | 01       | 11.78           | 10.23   | 9.8    |
> | 02       | 1.33            | 1.14    | 0.71    |
> | 03       | 72.51           | 71.53   | 71.1   |
> | 12       | **28.18**            | **10.51**   | **9.8**    |
> | 13       | 75.72           | 80.90  | 71.1   |
> | 23       | 75.55           | 71.81    | 71.1   |
> | 012      | 29.4            | 10.94    | 9.8    |
> | 013      | 76.8           | 81.33   | 71.1   |
> | 023      | 71.86           | 72.24   | 71.1   |
> | 123      | 73.4           | 81.61   | 71.1   |
>
> As the table shows, we observed:
>
> 1. Higher-order interactions do exist. For example, $E_{1}$ and $E_2$ in Cambrian-1 8B appear redundant individually (_CUR sum_ = $-0.53$), but their joint ablation leads to a $10.09\%$ performance drop—this confirms their complementary role that single-encoder masking fails to capture
> 2. Dominance of a key encoder. $E_3$ in Cambrian-1 8B and $E_3$ in Eagle-X4-8B act as a major contributor, all combinations include them show higher _CUR combination_ compared to those without it.
>
> So, our conclusion is for existing multi-encoder MLLMs with $>2$ encoders, higher-order interactions are still dominated by one specific encoder
>
> > Weakness 3: _Can the author provide the analysis of which fusion method is most effective and efficient? How can we decide which encoders are most important before training? Does the training strategy also influence the encoder selection? These are the follow-up questions. I do appreciate what the authors did in the paper._
>
> It is not simple to determine the most effective fusion mechanism since the final performance is determined by multiple factors. In practice, encoder importance is shaped jointly by:
>
> 1. The fusion mechanism
> 2. The training data distribution
> 3. The capacity of LLM
> 4. The optimization process, since encoder parameters are updated together with the LLM.
>
> We have considered comparing different fusion mechanisms, but a fair comparison is challenging due to inherent differences between methods. For example, the MLP used by Eagle and SVA used by Cambrian-1 has different number of parameters, MLP is a pre-fusion method (fusion occurs before entering into LLM), while SVA is a post-fusion method (fusion occurs after entering into LLM). But, our insight is simpler structure tends to have better results.
> The Eagle series is more robust than the Cambrian-1 series because Eagle’s fusion mechanism primarily 'passes through' visual information, while Cambrian-1’s mechanism involves additional 'processing' of visual features, by "passing" information, Eagle leave it to LLM to decide to use or drop information, thus making better use of visual information. Previous study such as Mousi[1] draws same conclusion where MLP achieves better performance than Q-former.
>
> **Again, we sincerely thank you for your valuable comments and suggestions, which have helped us substantially refine and clarify the paper. All newly added content has been incorporated into the latest version of the manuscript and highlighted in BLUE.**
>
> References:
>
> - [1] Fan, Xiaoran et al. “MouSi: Poly-Visual-Expert Vision-Language Models.” ArXiv abs/2401.17221 (2024): n. pag.

---

> ### Author Response · Authors · 2025-11-24
> **Short Summary of Discussion**
>
> Dear Reviewer rQtg, thank you again for your thoughtful review and constructive feedback. As we approach the end of the ICLR 2026 rebuttal period (and it has been a week since we posted our responses), we kindly ask you to review our detailed replies to your three concerns:
> **(1)** For the theoretical grounding of redundancy, we supplemented our empirical metrics with the *Platonic Representation Hypothesis* and new attention-map similarity analyses (showing high encoder alignment after masking non-essential encoders) to explain *why* redundancy arises and *how* feature spaces overlap.
> **(2)** For higher-order interactions, we conducted combinatorial CUR experiments confirming that while complementary effects exist (e.g., Cambrian-1’s encoders 1+2), performance in current multi-encoder MLLMs is often dominated by a single key encoder.
> **(3)** Regarding fusion methods and encoder selection, we clarified that effectiveness depends on multiple factors (fusion architecture, training data, LLM capacity), with simpler mechanisms (e.g., Eagle’s MLP) that minimally process visual features proving more robust than complex alternatives.
>
> All revisions are incorporated into the manuscript (highlighted in **blue**). **We sincerely hope these address your concerns, could you kindly confirm whether we’ve adequately resolved them, or if further clarifications would be helpful?** Thank you for your time and invaluable insights.

---

> > ### Comment · Reviewer_rQtg · 2025-11-25
> >
> > Thank you so much for your response! They are very helpful and I will make my score keep positive.

---

### Author Response · Authors · 2025-12-03
**Final Remark**

We sincerely thank the reviewers and Area Chairs for their insightful, constructive, and thorough feedback. We have carefully addressed all concerns raised by the reviewers.

Our work provides a **comprehensive empirical study on encoder redundancy in Multi-encoder Multimodal Large Language Models (MLLMs)**. We introduced **Conditional Utilization Rate (CUR)** and **Information Gap (IG)** as principled, model-agnostic metrics to systematically quantify encoder contribution and redundancy.

Our core finding is that simply adding more vision encoders often yields **severely diminishing returns and pervasive redundancy**, sometimes even harming performance. Notably, we show that dual-encoder variant can **recover over $90\\%$ of full performance** with **substantially lower training resources and inference latency**. This challenges the "more encoders are better" heuristic and provides actionable diagnostics for more efficient MLLM design.

In response to reviewer feedback, we have:

- **Strengthened Theoretical Grounding** (rQtg): We supplemented CUR/IG metrics with new analysis, including the Platonic Representation Hypothesis and attention-map similarity analyses, to explain the origins of redundancy via **representation convergence**. We clarified that CUR/IG provide a systematic, quantitative diagnostic framework.
- **Added Efficiency and Training Dynamics Analysis** (vJKH, yWcu, VGcs): We incorporated new experiments explicitly reporting **FLOPs, inference latency, and training time**. This data directly supports our claim that the marginal performance gains of full ensembles do not justify the substantial increase in required resources.
- **Quantified Higher-Order Interactions** (rQtg, VGcs): We introduced CUR combination metrics via combinatorial ablation, confirming that while complementary effects exist, performance in current architectures remains largely remains largely **dominated by a single key encoder**.
- **Clarified Future Work and Fusion** (rQtg, yWcu, VGcs): We positioned CUR/IG as foundational tools to guide future **learnable selection gating schemes** (like MoVA/MOVE). Our analysis also showed that redundancy persists across diverse fusion strategies (e.g., MLP vs. SVA).

We believe these revisions and additions have substantially refined and clarified the paper, solidifying its contribution to the community by providing a principled, quantitative framework for diagnosing and mitigating redundancy in multi-encoder MLLMs.

We appreciate the opportunity to engage with the community and look forward to the final decision.

---

### Meta-Review · Area_Chair_Zh6Q · 2026-01-06

**Summary:**

The paper investigates encoder redundancy in MLLMs using new metrics (CUR/IG). Primary reviewer concerns included theoretical grounding, higher-order encoder interactions, and training versus inference dynamics. The work is valued for converting qualitative intuitions into a rigorous, quantitative diagnostic framework despite remaining questions on training-time necessity.

**Reviewer Concerns:**

Rebuttals effectively addressed theoretical grounding, higher-order interactions, and efficiency metrics (FLOPs/latency). Concerns regarding "actionable" implementation of routing/pruning and the distinction between training-time necessity and inference-time redundancy remain partially outstanding, as authors positioned these as future work rather than diagnostic scope.

**Reviewer Scores:**

rQtg and VGcs would likely maintain or increase scores (7) following clarified interactions. vJKH and yWcu would likely move toward marginal acceptance (5 or 6), as their efficiency and coverage concerns were addressed.

---

### Decision · Program_Chairs · 2026-01-26

Accept (Poster)